# Genomic landscape associated with potential response to anti-CTLA-4 treatment in cancers

Chan-Young Ock[1,2], Jun-Eul Hwang[1,3], Bhumsuk Keam[2], Sang-Bae Kim[1], Jae-Jun Shim[1,4], Hee-Jin Jang [ID] [1,5], Sarang Park[1], Bo Hwa Sohn[1], Minse Cha[1], Jaffer A. Ajani[6], Scott Kopetz[6], Keun-Wook Lee[7], Tae Min Kim[2], Dae Seog Heo[2] & Ju-Seog Lee[1]

Immunotherapy has emerged as a promising anti-cancer treatment, however, little is known about the genetic characteristics that dictate response to immunotherapy. We develop a transcriptional predictor of immunotherapy response and assess its prediction in genomic data from ~10,000 human tissues across 30 different cancer types to estimate the potential response to immunotherapy. The integrative analysis reveals two distinct tumor types: the mutator type is positively associated with potential response to immunotherapy, whereas the chromosome-instable type is negatively associated with it. We identify somatic mutations and copy number alterations significantly associated with potential response to immunotherapy, in particular treatment with anti-CTLA-4 antibody. Our findings suggest that tumors may evolve through two different paths that would lead to marked differences in immunotherapy response as well as different strategies for evading immune surveillance. Our analysis provides resources to facilitate the discovery of predictive biomarkers for immunotherapy that could be tested in clinical trials.

[1] Department of Systems Biology, UT MD Anderson Cancer Center, Houston, TX 77054, USA. [2] Department of Internal Medicine, Seoul National University Hospital, Seoul 03080, Korea. [3] Department of Hematology-Oncology, Chonnam National University Medical School, Gwangju 61469, Korea. [4] Department of Internal Medicine, School of Medicine, Kyung Hee University, Seoul 02447, Korea. [5] Department of Molecular Oncology, The Graduate School of Medicine, Seoul National University, Seoul 08826, Korea. [6] Department of Gastrointestinal Medical Oncology, UT MD Anderson Cancer Center, Houston, TX 77054, USA. [7] Department of Internal Medicine, Seoul National University Bundang Hospital, Seongnam 13620, Korea. Correspondence and requests for materials should be addressed to J.-S.L. (email: jlee@mdanderson.org)

Understanding the interaction between cancer cells and the immune system has led to novel strategies for treating cancer[1–3]. The administration of tumor-infiltrating lymphocytes (TILs), interleukin-2, and vaccinations targeting tumor-specific antigens has prompted the treatment of cancer via host immune modulation[4, 5]. A recent strategy targeting immune checkpoints such as CTLA-4 and PD-1/PD-L1 has showed striking clinical benefit[6–8]. However, the overall response rates of advanced solid cancers to checkpoint inhibitors have been only modest (18–38%)[7, 8] with prolonged responses being even less common. Furthermore, marked response to immune checkpoint therapies have been limited to a subset of tumor lineages[9–11], suggesting that differences in organ physiology and molecular characteristics of various cancers may play a role in the efficacy of treatment response.

As seen in earlier studies demonstrating that therapeutic targets were reliable predictive biomarkers[12, 13], recent studies reported that tumor PD-L1 expression or its amplification was significantly associated with better response in patients undergoing anti-PD-1/PD-L1 therapies[11, 14, 15], although not all responders had high PD-L1 expression. Recent studies have shown that interferon-gamma target genes such as CXCL9, CXCL10, IDO1, IFNG, HLA-DRA, and STAT1 are indicative of response to immunotherapy in many cancers[16–19]. Moreover, TILs as well as PD-1 expression in TILs were also correlated with clinical outcomes[14], indicating that a better understanding of the immunologic landscape could lead to the identification of useful biomarkers for immunotherapy increasing the spectrum of patients able to benefit[20, 21]. Interestingly, recent small-scale genomic studies demonstrated significant correlation of mutational burden with response to immunotherapy[22, 23], suggesting that genomic alterations may dictate clinical outcomes of immunotherapies, as they do in targeted therapies. However, this contention has not been thoroughly tested in large cohorts of cancer patients across multiple cancer lineages.

In the current study, we aim to assess the potential benefit of immunotherapy across different cancer lineages and identify potential genetic markers associated with benefit of immunotherapy by developing a transcriptional profile from interventional studies integrated with unbiased systematic analysis of genomic data from The Cancer Genome Atlas (TCGA) project.

## Results

**Immune signature predicting response to immunotherapy.** Gene expression data from a randomized phase II trial of immunotherapy with MAGE-A3 antigen in malignant melanoma without prior treatment for metastases other than isolated limb perfusion were used for analysis[24, 25]. The tumor samples were obtained before the immunotherapy and clinical responders were defined by objective responders (complete and partial) according to RECIST 1.0[26] and patients showing stable disease (>4 months) or mixed response with unequivocal tumor shrinkage. In the current analysis, we identified 105 genes significantly associated with response to immunotherapy (P < 0.005 and 1.5-fold difference, Fig. 1a and Supplementary Data 1) and constructed a prediction model based on the Bayesian compound covariate predictor algorithm[27]. When patients were stratified according to Bayesian probability (cutoff = 0.5), responders were well separated from non-responders (AUC = 0.83, CI; 0.72–0.93, P < 0.001, Fig. 1d). We next sought to determine whether the predictor could also identify potential responders to different immunotherapy like anti-CTLA-4 antibodies. When applied to data from a mouse mesothelioma model treated with anti-CTLA-4 antibodies[28], our model reliably separated responders from

non-responders (AUC: 0.99, P < 0.001, 90% sensitivity, 90% specificity) (Fig. 1b, e). We next sought to determine if predictor can identify responders in clinical setting when applied to gene expression data from melanoma tissues of patients treated with ipilimumab[29]. Consistent with results from mouse model, our model reliably separated responders from non-responders (AUC: 0.7, P = 0.02) (Fig. 1c, f). Furthermore, patients classified as responders by predictor showed significantly favorable clinical outcome in both overall survival and progression-free survival (P = 0.009 and P = 0.03, respectively, Fig. 1g, h). Taken together, our data strongly suggest that the Bayesian probability of the immune signature (IS), hereafter referred to as the IS score, is associated with response to different immunotherapy approaches including MAGE-A3 antigen-based immunotherapy and anti-CTLA-4 immune checkpoint inhibitors. The prediction of responder by IS score has a good performance compared with other candidates of immune biomarker such as interferon-gamma signature[16, 17] or cytolytic activity[30] (Supplementary Figs. 1–3). IS score was not well associated with response to treatment with anti-PD-1 antibody in melanoma (N = 27) and renal cell carcinoma (N = 10), suggesting potential limitation of IS score predicting response to different immunotherapies (Supplementary Figs. 4, 5). However, it is worthwhile to point out that all other immune biomarkers failed to identify responders in these cohorts, indicating that lack of association might be due to small sample size. Pathway enrichment analysis of 105 genes showed activation of immune signaling pathways (Supplementary Fig. 6). In good agreement with predicted outcomes of anti-CTLA-4 antibody treatment, the CTLA-4 pathway was significantly activated, strongly supporting the notion that IS scores are associated with immunotherapy response at the biological and molecular levels. Consistent with pathway enrichment analysis, gene network analysis identified many pro-inflammatory cytokines and related transcription factors as potential upstream regulators activated in responder patients (Supplementary Data 2). On the contrary, anti-inflammatory cytokine IL10 and negative regulators of cytokine signaling such as SOCS1 and SOCS3 were activated in non-responder patients[31, 32] (Supplementary Fig. 7A). Interesting, same analysis revealed that MYC is activated in non-responders (Supplementary Fig. 7B). This is in good agreement with previous study demonstrating that MYC is negative regulator of immune response[33].

**Distribution of IS score in TCGA pan-cancer cohort.** Having found that the IS score reflected response to anti-CTLA-4 immunotherapies, we applied the IS to gene expression data from TCGA pan-cancer data including samples of 30 tumor types (N = 9081, Supplementary Data 3) to estimate the potential response rate of each cancer lineages to immunotherapy (Supplementary Fig. 8). As expected, cancers arising from lymphoproliferative tissues, such as diffuse large B-cell lymphoma, thymoma, and acute myeloid leukemia, had the highest IS scores, further supporting the notion that the signature reliably reflects immune activity in cancer tissues. When stratified into two subcategories (potential responder: >0.5 and non-responder: <0.5), kidney clear cell carcinoma (KIRC), lung adenocarcinoma (LUAD), and cervical and endocervical cancer (CESC) had the highest median IS scores indicative of large proportion of potential responders (Fig. 2a, b). The proportion of potential responders to immunotherapy highly varied within each type of solid cancer ranging from 0.5 to 65.9%. Interestingly, among the solid cancers, skin cutaneous melanoma (SKCM) had a relatively high proportion of predicted responders (33.7%) even though the median IS score was not top-ranked, because IS scores in SKCM were skewed to a high level.

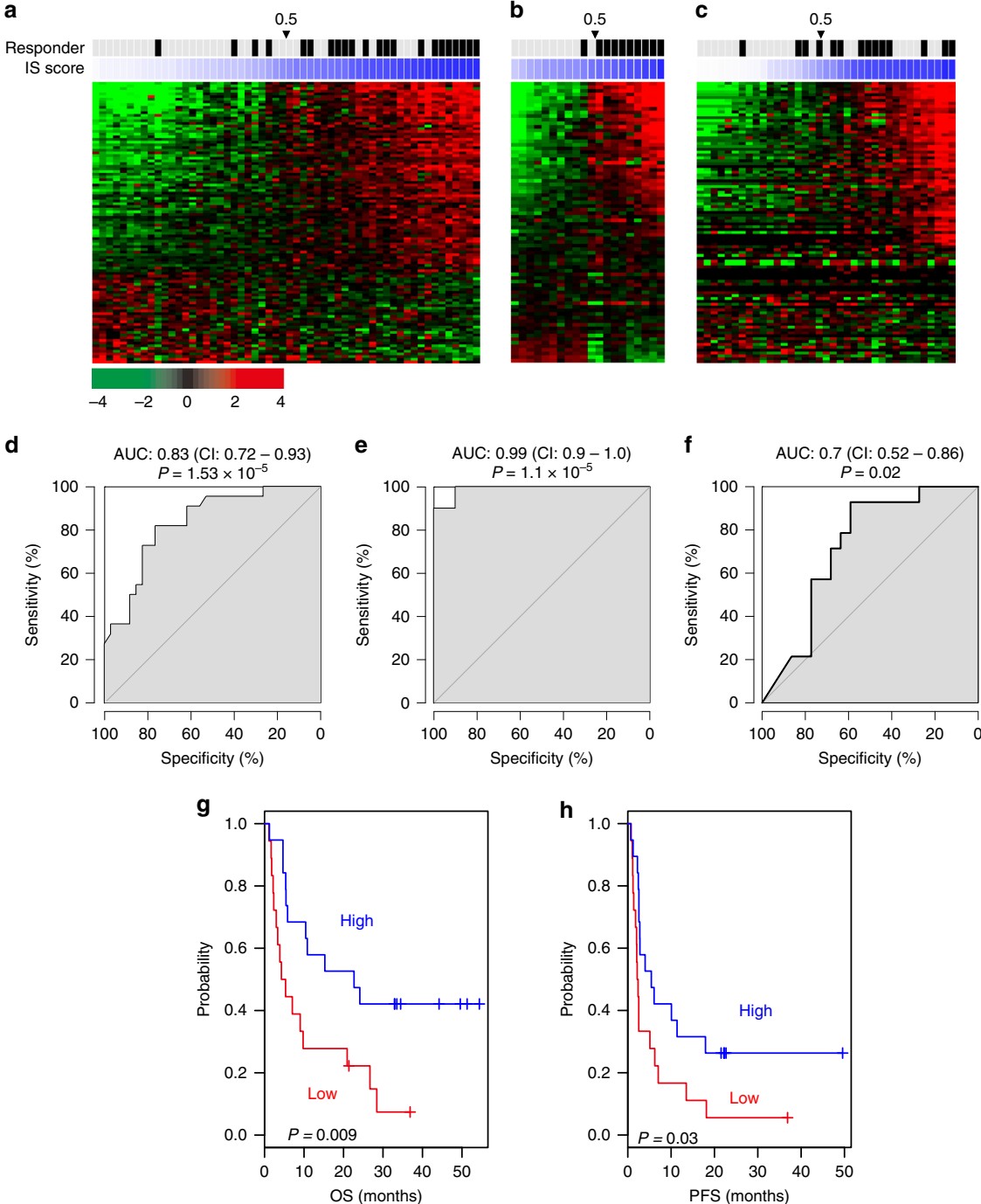

**Fig. 1** Immune signature reflecting response to immunotherapy from human and mouse cancer tissues. **a** Expression patterns of genes significantly associated with response to immunotherapy in training cohort. Pretreatment biopsies from patients with metastatic melanoma were used to generate gene expression data. Genes whose expression is significantly different between responders and non-responders were selected (105 genes, $P < 0.005$ and 1.5-fold difference). The data are presented in matrix format, with rows representing the individual gene and columns representing each tissue. Each cell in the matrix represents the expression level of a gene feature in an individual tissue. Red and green reflect high-expression and low-expression levels, respectively, as indicated in the scale bar (log 2 transformed scale). Immune signature (IS) scores are presented as color index in blue (0−1 scale). **b** Expression patterns of immune signature genes and IS scores from mouse mesothelioma model treated with anti-CTLA-4 antibodies. **c** Expression patterns of immune signature genes and IS scores from human melanoma tissues treated with ipilimumab. **d**–**f** Receiver operating characteristics (ROC) analysis of IS scores from each prediction. Robustness of IS scores identifying responders to immunotherapy was estimated by area under curve (AUC) from ROC analysis. **d** Human melanoma treated with MAGE-A3 antigen, **e** mouse mesothelioma model treated with anti-CTLA-4 antibodies, **f** human melanoma treated with ipilimumab. (CI: 95% confident internal of AUC). **g**, **h** Kaplan−Meier plots of overall survival (OS) and progression-free survival (PFS) of advanced melanoma patients treated with ipilimumab. Patients were stratified according to IS scores (high >0.5). See also Supplementary Figs. 1 −7 and Supplementary Data 1, 2

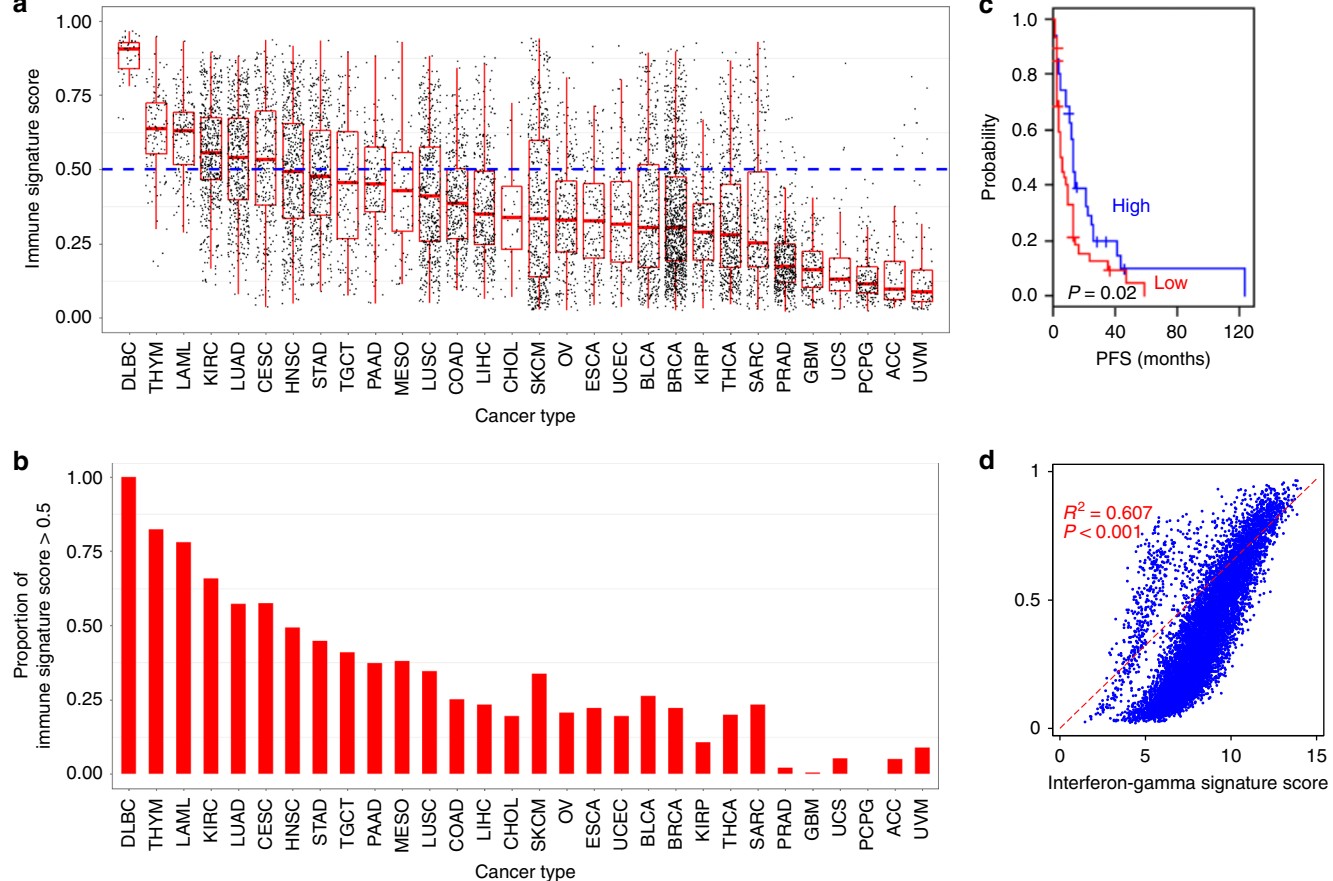

**Fig. 2** Immune signature scores and potential responders to immunotherapy across major cancer types. **a** Immune signature (IS) score calculated by Bayesian probability of immune signature is plotted according to cancer types. Each black dot represents IS score, and red lines in the box represent upper 75%, median, and lower 25% values of each cancer type. Blue line represents IS score of 0.5. **b** Proportion of potential responders to immunotherapy (IS score > 0.5) is shown according to cancer types. **c** Kaplan–Meier plots of progression-free survival (PFS) of advanced melanoma patients in TCGA treated with immunotherapy. Of 472 patients with melanoma, only 78 patients treated with immunotherapy were included in analysis. Patients were stratified according to IS scores (high > 0.5). **d** Association between IS score and interferon-gamma signature score. Scatter plot and fitted dash line showed the significant association between IS score and interferon-gamma signature score. Abbreviation of cancer type was referred from The Cancer Genome Atlas tag. See also Supplementary Figs. 8–11 and Supplementary Data 3, 4.

IS score was significantly correlated with progression-free survival of 78 SKCM patients who were received immunotherapy in TCGA ($P = 0.024$, Fig. 2c and Supplementary Data 4). Moreover, IS score was significantly correlated with interferon-gamma score ($R^2 = 0.607$, $P < 0.001$), which can predict the responders of anti-PD-1 antibody in the previous studies[16, 17]. PD-L1 mRNA expression and PD-1 mRNA expression, which were also proposed to be related with the responder of anti-PD-1/PD-L1 inhibition[11, 14, 21], were significantly correlated with IS scores (Supplementary Fig. 9) even though these are not components of the IS score, further supporting the notion that IS scores reflect underlying biology that determines the outcomes of immunotherapy.

Consistent with previous indications that patients with immunogenic tumors had a favorable survival outcome[34], patients with high IS scores (>0.5) showed significantly favorable overall survival in bladder cancer (BLCA) although they were not treated for immunotherapy (Supplementary Fig. 10).

For estimation of relative fractions of immune cells in each tumor, we used CIBERSORT to infer relative RNA fractions of 22 different immune cells[35]. Not surprisingly, fraction of CD8+ T cells and M1 macrophages were most significantly associated with IS scores (Supplementary Fig. 11), further supporting that IS scores well reflect infiltrated active immune cells in tumor mass.

**Association of IS scores with molecular subtypes of cancers.** We next assessed the association of IS scores with molecular subtypes defined by TCGA studies[36–43]. In SKCM[36], although IS scores were only modestly associated with four mutation subtypes, they were significantly associated with platform subtypes (Supplementary Fig. 12). IS scores were significantly higher in the immune-high mRNA subtype and normal-like methylation subtype. When tumors in the immune-high mRNA subtype were further stratified by methylation subtype, IS scores were significantly higher in the normal-like subtype than in other subtypes ($P = 4.1 \times 10^{-8}$, Fig. 3a). Interestingly, IS scores were lower in the RAS subtype than in other subtype (Supplementary Fig. 12).

A TCGA study classified thyroid cancer (THCA) into BRAF-like and RAS-like subtypes[37]. Consistent with observations in SKCM, IS scores were significantly lower in the RAS-like subtype than in the BRAF-like subtype (Supplementary Fig. 13, $P = 3.5 \times 10^{-19}$). Among platform subtypes in THCA, the C1 methylation subtype was most significantly associated with IS scores, whereas the distribution of IS scores was skewed toward high in the follicular methylation subtype (Supplementary Fig. 13). When BRAF-like subtypes were further stratified according to methylation subtype, C1 and follicular subtypes were more significantly associated with higher IS scores than other methylation subtypes ($P = 1.2 \times 10^{-30}$, Fig. 3b). In head and neck squamous cell

carcinoma (HNSC), most of the molecular subtypes were significantly associated with IS scores (Supplementary Fig. 14). IS scores were significantly higher in the C3 copy number alteration (CNA) subtype, hypermethylation subtype, mesenchymal mRNA subtype, and C3 miRNA subtype than in all the other subtypes. This association was independent of human

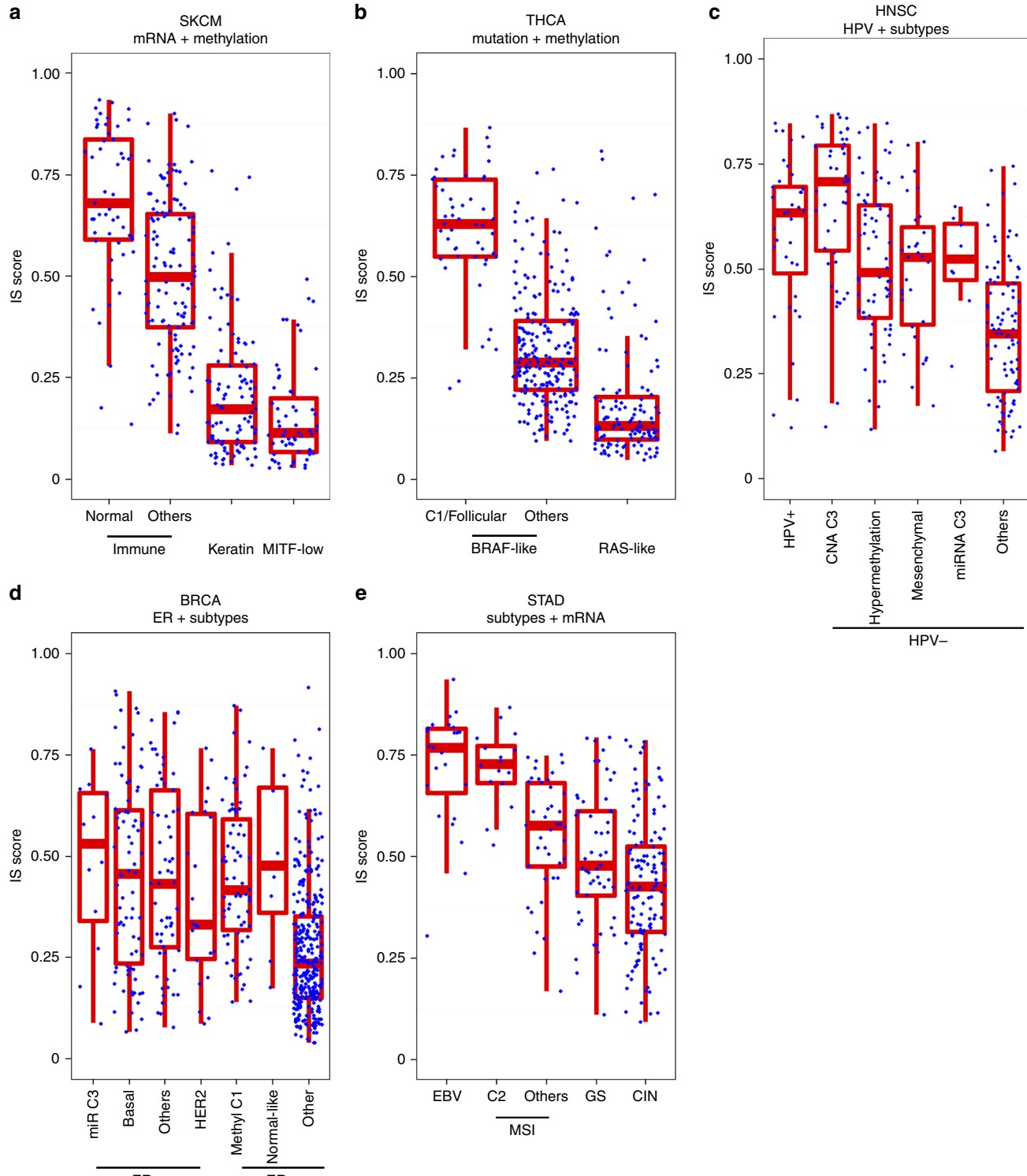

**Fig. 3** Molecular subtypes of cancers associated with potential response to immunotherapy. **a** Melanoma tumors were grouped according to mRNA subtypes first and immune-high subtype was further stratified by methylation subtypes (Normal-like or others). **b** Thyroid tumors were grouped according to mutation subtypes first and BRAF-like subtype was further stratified by methylation subtypes (C1/Follicular or Others). **c** Head and neck squamous cell carcinoma tumors were grouped by HPV-infection status first and HPV-negative tumors were further stratified by molecular subtypes. **d** Breast tumors were grouped by ER status first and tumors were further stratified by molecular subtypes. **e** Stomach tumors were grouped by molecular subtypes first and MSI tumors were further stratified by mRNA subtype (C2 or others). Blue dot represents IS score, and red lines in the box represent upper 75%, median, and lower 25% values of molecular subtype. See also Supplementary Figs. 12–19

papillomavirus (HPV) status because IS scores remained high in these subtypes when HPV-negative tumors only were analyzed (Supplementary Fig. 15). IS scores in each sensitivity subtype remained high when HPV-negative tumors were subsequently stratified into different subtypes (Fig. 3c), suggesting that molecular mechanisms driving sensitivity to immunotherapy might be different in each sensitive subtype.

A TCGA study revealed a similarity between lung squamous cell carcinoma (LUSC) and HNSC[38, 39]. In good agreement with this, IS scores were significantly higher in secretory mRNA subtypes of LUSC that is highly related to mesenchymal mRNA subtypes in HNSC (Supplementary Fig. 16, top). In LUAD[40], IS scores were significantly higher in the proximal inflammation mRNA subtype and CIMP-intermediate methylation subtype than in other subtypes (Supplementary Fig. 16, bottom). In BLCA[41], IS scores were highest in the infiltrated/mesenchymal mRNA subtype (Supplementary Fig. 17), suggesting potential role of signaling events governing epithelial to mesenchymal transition in cancer immunity. In BRCA[42], IS scores were significantly higher in estrogen receptor (ER)-negative tumors (Supplementary Fig. 18). When the BRCA subtype was further stratified, HER2 mRNA subtype had the lowest IS scores among ER-negative tumors, whereas the C1 methylation subtype and normal-like mRNA subtype had higher IS scores than other ER-positive tumors (Fig. 3d). The majority of the ER-positive showed much lower IS scores, indicating that in addition to genomic and epigenetic alterations, ER is a major determinant of cancer immunity in BRCA. In stomach adenocarcinoma (STAD)[43], the Epstein-Barr virus (EBV) subtype had the highest IS scores among four molecular subtypes (Supplementary Fig. 19). In the microsatellite instability (MSI) subtype, a substantial number of tumors were the C2 mRNA subtype and C2 subtype had significantly higher IS scores than others (Fig. 3e). Most interestingly, subtypes with higher IS scores in different cancers were associated with low genomic CNAs (i.e., the C6 iCluster subtype of LUAD, C3 CNA subtype of HNSC, C1 CNA subtype of BRCA, and low CNA subtype of STAD), indicating a potential connection of genomic instability to tumor immunogenicity that may govern clinical outcomes of immunotherapies.

**Association of IS scores with genomic characteristics of cancer.** Since previous small-scale clinical studies indicated a potential association of mutation burdens with immunotherapy response[22, 23], we tested the association of IS scores with mutations rates in TCGA data set ($N = 6162$) (Supplementary Fig. 20, top). The number of predicted neoantigens[30] was significantly associated with the mutation rates regardless of mutation types (Supplementary Fig. 20, bottom). Interestingly, a global analysis of all tumors showed a significant positive correlation between the non-synonymous mutation rate and the IS score ($R^2 = 0.017$, $P < 0.001$, Supplementary Fig. 21, top). In particular, this association was more significant in colorectal adenocarcinoma (COAD), STAD, and BRCA (Fig. 4a and Supplementary Fig. 22, top).

Because our analysis indicated a potential association of chromosomal instability (CIN) with IS scores, we assessed the global association of CIN with IS scores ($N = 8637$) by generating a "CIN score"[44]. As expected, CIN scores accurately reflected the overall CIN of tumors (Supplementary Fig. 23). Most interestingly, CIN scores had a significant negative correlation with IS scores ($R^2 = 0.095$, $P < 0.001$, Supplementary Fig. 21, bottom) to a greater degree than mutation rates. Furthermore, the trends of negative correlation were observed in most cancers (Fig. 4b and Supplementary Fig. 22, bottom), strongly suggesting that CIN might be a more important predictor of clinical outcomes of immunotherapy than mutation rates.

Since our results revealed a correlation of two genomic alterations types with IS scores, we next integrated non-synonymous mutation rates with CIN scores ($N = 5989$) to assess the interplay of two genomic alterations in cancer immunity. When two data sets were integrated, tumors were clearly separated into three major groups: tumors with high mutational burden and low CIN (mutator or M type), those with low mutational burden but high CIN (chromosome-instable or C type), and those not otherwise specified (NOS) (Fig. 4c). Consistent with previous observation, M-type tumors had high IS scores, whereas C-type tumors had low IS scores. IS scores were significantly higher in MSI-high tumors than in MSI-low or microsatellite-stable tumors (Supplementary Fig. 24, top), consistent with MSI tumors having high mutation rates and relatively low CNAs (Supplementary Fig. 24, bottom) as well as markedly increased responses to anti-PD-1 immunotherapy[23]. Furthermore, the proportion of M type was well correlated with IS score in each cancer type with the exception of KIRC (Fig. 5). Although MSI-H tumors have highest average IS scores among MSI subtypes, some of them have much higher IS scores, suggesting additional layer of regulatory mechanisms. Further analysis of gene expression data from MSI-H subtypes indicate that several interleukins (*IL4*, *IL15*, and *IL21*) are more active in tumors with high IS scores (Supplementary Fig. 25).

Because CNA can be influenced by tumor purity in tumor tissues[45], we estimated the potential impact of tumor purity in our analysis by examining the correlation of CIN scores with histologically assessed tumor purity. The correlation between CIN scores and tumor purity was only modest (Supplementary Fig. 26, top). Interestingly, non-synonymous mutation rates were also modestly correlated with tumor purity, suggesting that the correlation was not specific to CIN scores. Furthermore, the significance is not markedly altered by reanalysis of integrated data with adjusted CIN scores (Supplementary Fig. 26, bottom), strongly indicating a minimum impact of tumor purity in our analysis. To further validate insignificant contribution of tumor purity to CIN and IS scores, we adopted previously established genomic approach, consensus measurement of purity estimations (CPE)[46], for estimation of tumor purity that use gene expression, copy number alterations, and methylation data. As seen with IHC data, the correlation between CIN scores and tumor purity was modest (Supplementary Fig. 27, top) and the significance is not markedly altered by reanalysis of integrated data with adjusted CIN scores (Supplementary Fig. 27, bottom). Not surprisingly, IS scores are positively correlated with high stromal fraction in tumor mass (Supplementary Fig. 28), probably reflecting higher infiltration of immune cells.

**Somatic mutations positively associated with IS scores.** We next examined the association of IS scores with somatic mutations in 373 genes that have been designated drivers in previous studies[47] (Fig. 6a and Supplementary Data 5, 6). Strikingly, the majority of the significantly mutated genes were positively correlated with IS scores, suggesting that some of them might contribute host immunity. Interestingly, three of the significant genes were *MUC4*, *MUC17*, and *MUC7*, members of the mucin family that were previously identified as tumor antigens[48–50], and this finding supports our hypothesis. In contrast to tumor antigens, some of the positively correlated mutations might be selected under host immunogenic pressure as part of cancer cells' mechanisms to evade immune surveillance. Mutations in *HLA-A*, *-B*, and *-C*, *B2M*, and *CASP8* might fall into this category since *CASP8* is an executor of ligand-mediated apoptosis[51] and *HLA-A*, *-B*, and *-C* and *B2M* encode major antigen presenting machinery to immune cells[52]. Any loss-of-function mutations would give a significant

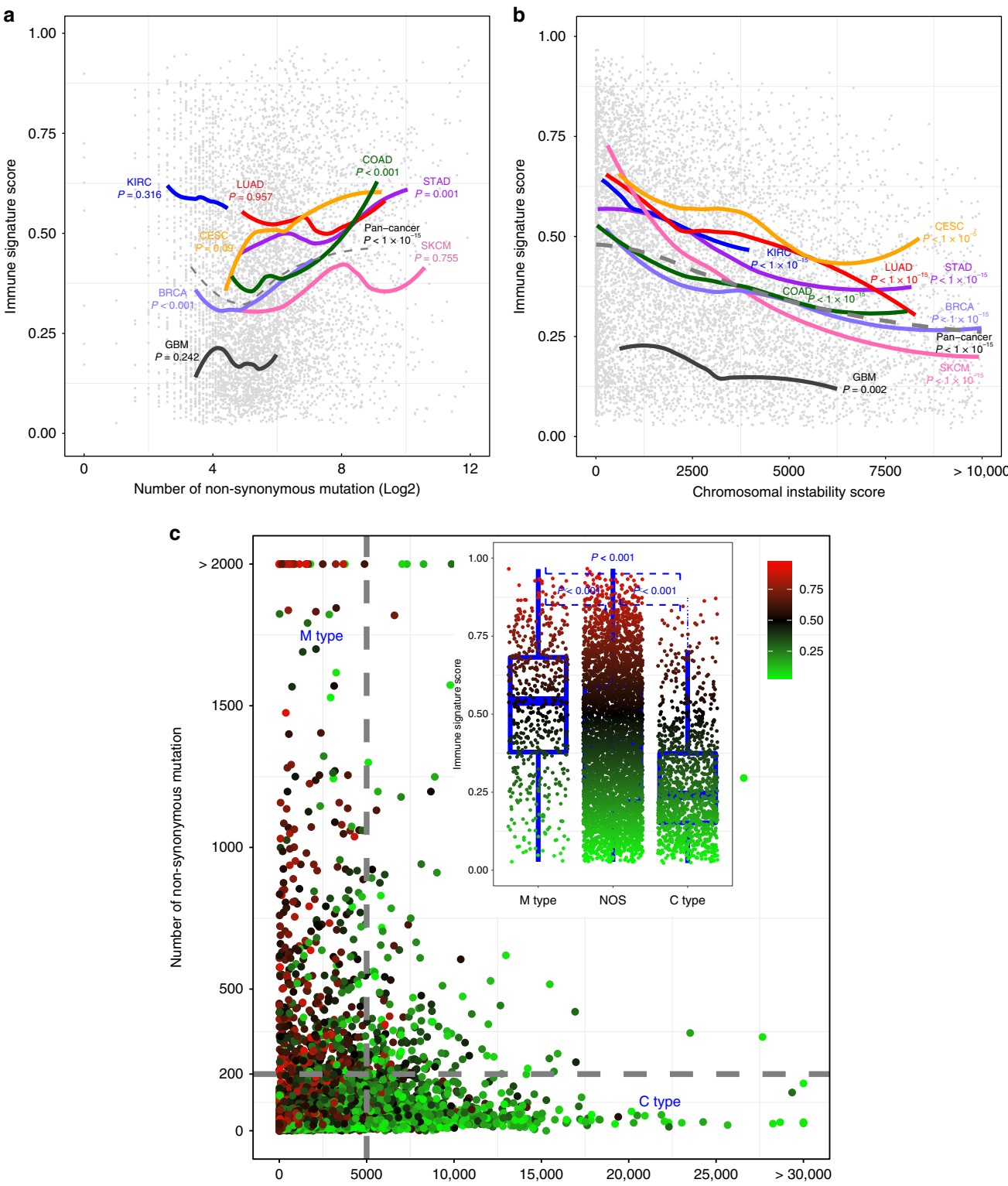

**Fig. 4** Two types of tumors distinct in genomic alterations and response to immunotherapy. **a** Scatter plots between mutation rates and IS scores. Dotted line represents global regression curve between mutation rates and IS scores in all cancers. Solid lines represent local regression curves between mutation rates and IS scores in each cancer as indicated. **b** Scatter plots between chromosome instability (CIN) scores and IS scores. Dotted line represents global regression curve between two scores in all cancers. Solid lines represent local regression curves between two scores in each cancer as indicated. CIN scores were defined by sum of square of gene-level gistic 2 value. **c** Scatter plots between mutation rates and CIN scores. Size and colors of dots represents IS scores as indicated in reference index. The tumor with high mutational burden (M type) is defined by number of non-synonymous mutation more than 200 as described in previous study, whereas the tumor with high chromosomal instability (C-type) is defined by CIN score more than 5000. Otherwise, tumors are classified as not otherwise specified (NOS). IS score according to M type or C type is summarized inside the graph. Blue lines in the box represent upper 75%, median, and lower 25% values of subtype. See also Supplementary Figs. 20–28

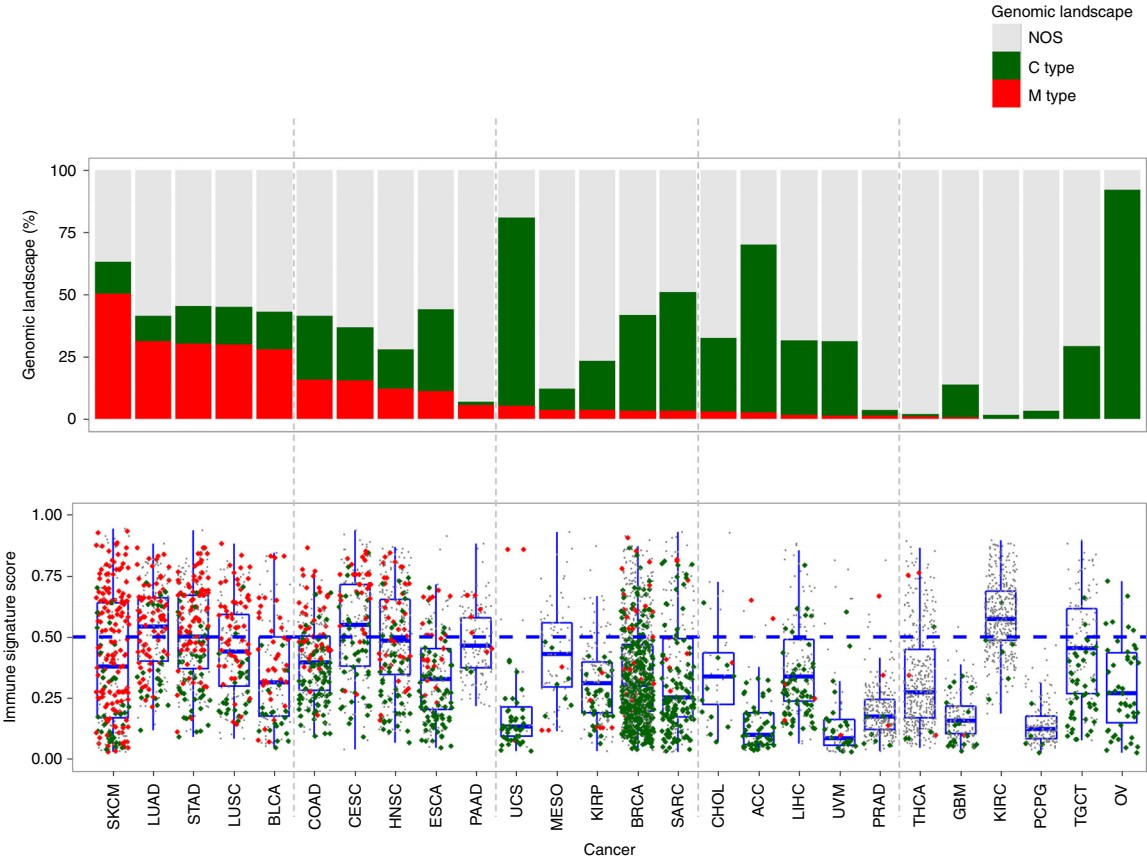

**Fig. 5** Distribution of genomic cancer types in each cancer. (top) A systematic overview of proportion of mutator type tumors (M type, red), chromosome-instable tumors (C type, green), and not otherwise specified (NOS) (bottom) immune signature (IS) score in each cancer. Blue lines in the box represent upper 75%, median, and lower 25% values of cancer type. Abbreviation of cancer type was referred from The Cancer Genome Atlas tag name

advantage to cancer cells to evade immune surveillance. In good agreement, tumors with mutations in these genes represent typical M type characteristics (Fig. 6b–d).

**Copy number alteration negatively associated with IS scores.** We next examined the association of IS scores with previously identified copy number-dependent drivers (87 amplification and 123 deletion)[47, 53] (Supplementary Fig. 29, Supplementary Data 5, 7, 8). In contrast to mutations, the majority of the significantly amplified genes were negatively correlated with IS scores (Fig. 7a). Likewise, the majority of the deleted genes also had a significant negative association with IS scores (Fig. 7b), suggesting that this type of genetic event is not prone to stimulate host immunity and that activated or suppressed genes may play a role in the suppression of host immunity. Consistent with our observations in BRCA (Fig. 3d), the amplification of *ERBB2* (*HER2*) was significantly associated with low IS scores. Amplified genes negatively associated IS scores include well-known driver oncogenes such as *MYC* and *E2F3* while deleted genes with negative association include well-known tumor suppressor genes such as *RB1*, *TP53*, and *PTEN*. Importantly, recent study demonstrated that loss of *PTEN* is indeed significantly associated with resistant to immunotherapy with anti-PD-1 antibodies in melanoma[54], strongly suggesting that many of identified candidates might play key roles in host immunity to cancer cells. Interestingly, expression of *HLA-A*, *HLA-B*, and *HLA-C* had a significant negative correlation with CIN scores in tumors with amplified genes or deleted genes (Fig. 7c), suggesting that some of the copy number-altered genes might be involved in the suppression of antigen presentation in cancer cells either alone or in combination with

other genes. Further supporting this notion, the expression of HLA genes was further reduced in tumors with co-amplified *MYC* and *FGFR1* (Fig. 7c).

**Association of IS scores with viral presence.** Not surprisingly, EBV-positive STAD and HPV-positive HNSC tumors were significantly associated with higher IS scores ($P < 0.001$, Supplementary Fig. 30). However, hepatitis B virus positivity was not associated with IS scores in liver hepatocellular carcinoma (LIHC), or other cancers (Supplementary Fig. 30, bottom right).

**Discussion**
In the current study, we generated IS scores based on response to different immunotherapy approaches in patients and in a model system and applied them to major cancer types. The analysis revealed two distinct types of tumors (M type and C type) that differ in their potential response to immunotherapy. Our analysis suggested that tumors evolve through two major paths that have different mechanisms for activating driver genes and may account for difference in immunotherapy response as well as strategies for evading immune surveillance.

While initially uncovered by analyzing the data from a vaccine immunotherapy approach, several lines of evidences strongly support that IS and IS scores are applicable to other types of immunotherapy. First, IS scores reliably identified responders to immunotherapy in a mouse model treated with anti-CTLA-4 antibodies. Second, pathway enrichment analysis identified the CTLA-4 pathway as one of the key pathways activated in the signature. Further supporting this finding, the iCOS-iCOSL pathway that is activated in the signature was recently

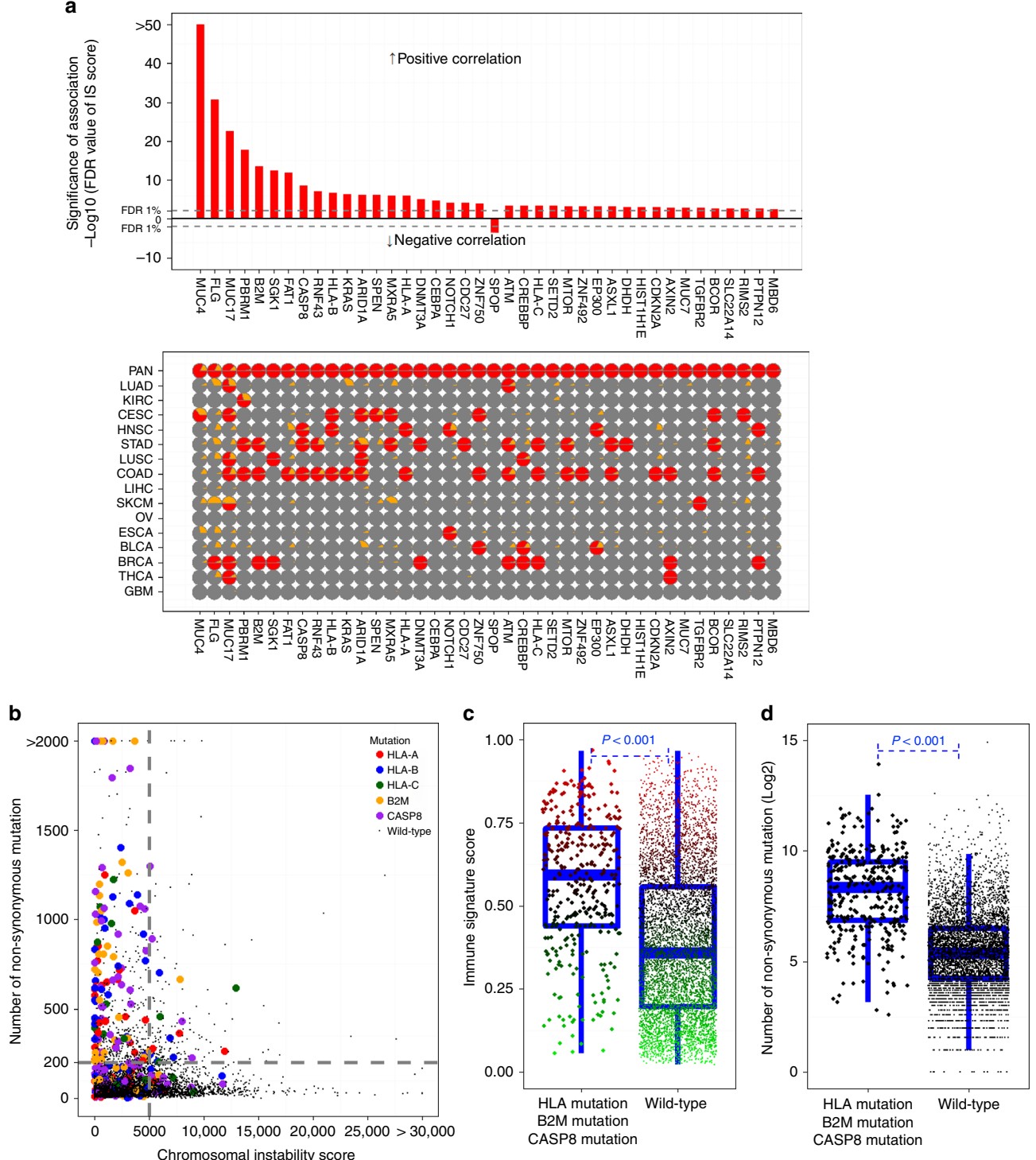

**Fig. 6** Somatic mutations associated with immune signature scores. **a** Top: the significant pan-cancer-wide association of mutation rates with immune signature (IS) scores. Red bars represent significance (FDR) of association of mutation rate with IS score in each gene. Dashed line indicates 1% threshold of FDR. Bottom: the significance of association in each cancer type. Red or gray indicate significance or insignificance, respectively, and yellow indicates the frequency of mutations in each gene. **b** Scatter plots between mutation rates and chromosomal instability scores in all tumors highlighted with mutations in HLA-A, HLA-B, HLA-C, B2M, and CASP8. **c**, **d** IS score **c** and number of mutation **d** in tumors with and without mutations in HLA-A, HLA-B, HLA-C, B2M, and CASP8. Blue lines in the box represent upper 75%, median, and lower 25% values of subtype. See also Supplementary Data 5

identified as a pharmacodynamic marker for anti-CTLA-4 therapy[55]. Third, most importantly, IS scores can identify responder patients with melanoma after treatment with ipilimumab[29]. Forth, IS scores is significantly correlated with interferon-gamma score that is predictive markers for anti-PD-1 therapy in gastric

and head and neck cancer. Furthermore, IS scores were significantly associated with expression of PD-1 and PD-L1 in TCGA data. Fifth, gene network analysis identified many pro-inflammatory cytokines as activated upstream regulators in responder patients while it identified anti-inflammatory cytokines

and negative regulators of cytokine signaling as activated regulators in non-responder patients. Moreover, it also identified MYC as negative regulator of immune activity. Indeed, recent

study demonstrated that MYC is negative regulator of immune[33]. Finally, IS scores predicted that MSI tumors would have strong responses to immunotherapy which is supported by clinical trial

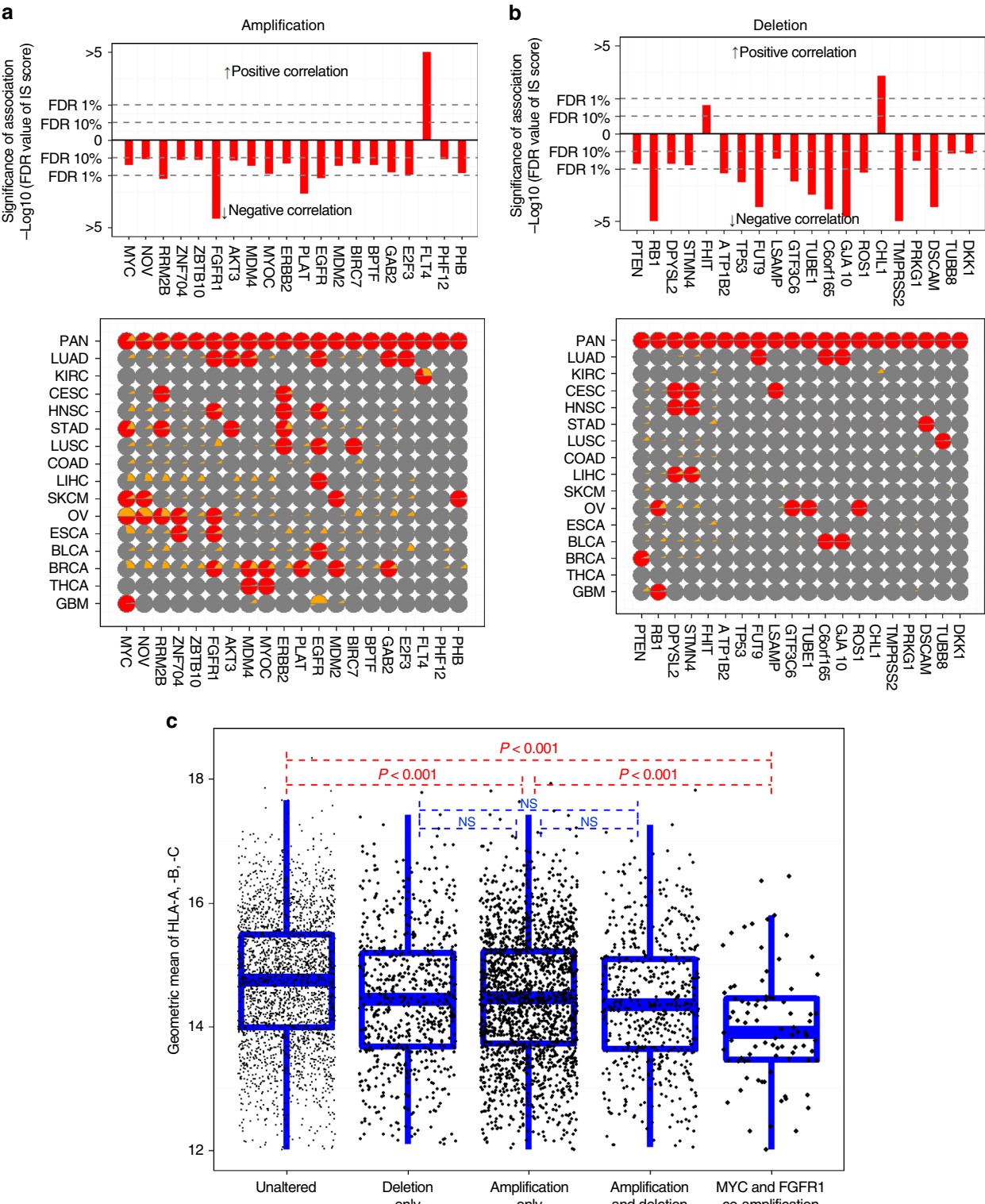

**Fig. 7** Somatic copy number alterations associated with immune signature scores. **a**, **b** Top: the significant pan-cancer-wide association of amplified **a** and deleted **b** genes with immune signature (IS) scores. Red bars represent significance (FDR) of association of amplification with IS scores in each gene. Bottom: The significance of association in each cancer type. Red or gray indicate significance or insignificance, respectively, and yellow indicates the frequency of amplification in each gene. **c** Mean expression of HLA-A, HLA-B, and HLA-C in not significantly altered, deleted only, amplified only, both amplified and deleted, and *MYC* and *FGFR1* co-amplified tissues. Blue lines in the box represent upper 75%, median, and lower 25% values of subtype. See also Supplementary Fig. 29 and Supplementary Data 6, 7

data[23]. Taken together, these observations strongly suggest that IS scores well reflect underlying biology that may play key roles in clinical outcomes.

Although immunotherapy has led to great enthusiasm for treatment of a subset of cancer types, including melanoma, non-small cell lung cancer, and kidney cancer[56], its clinical effects have been disappointing in other tumor lineages. While recent studies using genomic approaches have begun to shed light on genomic alterations associated with the benefits of immunotherapy[21, 22, 57], the underlying biology of predicting benefit of immunotherapy is poorly understood. Systematic integration of somatic mutations and CNAs in connection with our IS predictor of response to immunotherapy revealed two distinct types of tumors (Supplementary Fig. 31). M-type tumors are rich in somatic mutations, low in CNAs, and likely to be sensitive to immunotherapy. Some of mutated gene products such as mucins may provide highly immunogenic antigens and may be accountable for high IS scores. In contrast to M-type tumors, C-type tumors are high in CNAs, low in mutations, and likely to be resistant to immunotherapy. This finding is in good agreement with recent study showing that high copy number alteration is potential predictive marker for immunotherapy[58]. In another study analyzing of samples from clinical trials with CTLA-4 and PD-1 blockade treatments, copy number loss is associated with resistance to immunotherapy[59]. Molecular mechanisms of resistance to immunotherapy is currently unknown. A lack of neoantigen production due to low mutation rates may account for the insensitivity of C-type tumors to immunotherapy, however loss of key immune mediators is also likely to contribute.

Evasion of immune surveillance is necessary for cancer cells to survive and grow[60]. Two tumor types may adopt different strategies to evade immune surveillance. M-type tumors have frequent mutations in genes involved in antigen presentation. Mutations in HLA-A, -B, and -C and B2M might arise under selective pressure to evade host immunity. Likewise, mutations in CASP8, which is a key mediator of apoptosis[51], give an advantage to cancer cells to become insensitive to T-cell-mediated cell death. Similar findings were also observed in previous study using immune cytolytic score as immune activity in tumor mass[30]. While interesting, these associations should be interpreted with caution and need to be validated in prospective studies.

Amplified and deleted genes in C-type tumors are significantly associated with lower expression of HLA-A, HLA-B, and HLA-C genes, suggesting that they may suppress the expression of these antigen-resenting genes to evade immune surveillance and may account for low IS scores in C-type tumors. In good agreement with our analysis that identified PTEN as a key modulator of tumor immunity, recent study showed that PTEN play roles in T-cell activation and loss of PTEN is significantly associated with resistance of melanoma to immunotherapy[54]. Likewise, MYC was predicted to be negative regulator of tumor immunity. Recent study also demonstrated that MYC inhibits T-cell activation by upregulating CD47 and PD-L1[61], further supporting validity of our approaches. Therefore, it is important to determine in future experiments whether other amplified or deleted genes are secondary therapeutic targets that can improve the efficacy of immunotherapy. The proportions of M-type tumors were generally well correlated with IS score in many cancer types. However, KIRC tumors had very high IS scores and proportion of M-type tumors was low (Fig. 5), suggesting that a high mutation rate does not fully account for IS scores.

IS scores are clearly associated with clinical subtypes of cancers. As expected, tumors with viral infection had high likelihood of response to immunotherapy. Our analysis also showed that the benefits of immunotherapy may not be limited to viral infected tumors. In HNSC, IS scores in the C3 CNA and mesenchymal mRNA subtypes were higher than or almost equal to IS scores in HPV-positive tumors. In STAD, the C2 mRNA subtype had equally high IS scores

with EBV-positive tumors. Furthermore, a subset analysis of SKCM, THCA, and BRCA tumors showed a significant difference in IS scores among clinical subtypes. In good agreement with a previous study showing a high response rate of basal type breast cancer (24%) to anti-PD-L1 antibody[62], the basal subtype had high IS scores in our analysis. These results indicate that subtype-specific biomarkers would improve the efficacy of immunotherapy in future trials.

The results of our study should be further validated in a prospective cohort of patients receiving immunotherapy. Although IS scores were validated in melanoma patients treated with anti-CTLA-4 antibodies, we cannot rule out the possibility that IS scores are more specific to tumor vaccines and probably to melanoma. Moreover, our result should be interpreted carefully when it applied to other cancer types as IS score is mostly validated in melanoma. Differences in genetic makeup of cancer cells and tumor microenvironment might have substantial influence on IS score in other cancer types. This should be further tested and validated in future studies with data from prospectively collected samples. As not all patients with high IS scores have greater benefit of immunotherapy, more clinical factors should be incorporated to prediction models for improvement of accuracy. As tumor tissues in TCGA are relatively in the early stages, our results should be interpreted with caution since later stages of tumors may have different composition of immune cells.

In the current study, we showed that the potential benefit of immunotherapy highly varies across cancer lineages and revealed global subtypes of tumors and genomic alterations significantly associated with the potential benefit of immunotherapy. Our findings could lead to opportunities to discover new biomarkers for immunotherapy that can identify subsets of patients who could derive greater benefit from immunotherapy.

## Methods

**Genomic and clinical data sets.** We used publicly available data in the current study. Gene expression data used for identification of IS and generation of IS score (accession number GSE35640[24]), and validation of IS scores in mouse model treated with anti-CTLA-4 antibody (accession number GSE63557[28]), in human melanoma patients treated with anti-PD-1 antibody (accession number GSE78220[63]), human renal cell carcinoma patients treated with anti-PD-1 antibody (accession number: GSE67501[64]) were obtained from Gene Expression Omnibus database (http://www.ncbi.nlm.nih.gov/geo). Another data set of RNA expressions regarding the validation of IS scores in human melanoma treated with anti-CTLA-4 antibody[29] was generously given by the authors (Van Allen EM and Garraway LA). All other data from TCGA project were obtained from TCGA data portal (https://tcga-data.nci.nih.gov) and cancer browser (https://genome-cancer.ucsc.edu). Gene-level gene expression data from RNA-seq experiments ($N = 9081$), copy number variation data ($N = 8785$), tumor purity data ($N = 8149$), somatic mutation data ($N = 6162$), clinical information data (overall survival, $N = 8522$), and microsatellite status of tumors ($N = 1103$) were included in analyses. Among TCGA data set, we excluded data for brain lower grade glioma due to indolent behavior and kidney chromophobe renal cell carcinoma due to rare incidence and far different tumor biology to other renal cell carcinoma. Altogether, samples of 30 major cancer types ($N = 9081$) were included in the final analysis (Supplementary Data 3). Somatic mutation data of HLA-A, HLA-B, and HLA-C genes were obtained from previous study that used the algorithm Polysolver[65], HLA somatic mutations were available in 6162 patients for our analysis. Viral presence status and number of predicted neoantigen of TCGA samples ($N = 3658$) was obtained from a previous publication[30]. Genetic and molecular subtypes of skin cutaneous melanoma, thyroid cancer, head and neck squamous cell carcinoma, breast cancer, stomach adenocarcinoma, lung adenocarcinoma, lung squamous cell carcinoma, and bladder urothelial cell carcinoma were obtained from previous TCGA publications[36–43]. Of 472 patients with skin cutaneous melanoma, 78 patients treated with immunotherapy which purpose is not indicated by "adjuvant" with appropriately annotated survival data were included in progression-free survival analysis.

**Analysis of the data, IS scores, and CIN scores.** For the number of total somatic mutations, multiple somatic mutations including non-synonymous mutation, insertion-deletion mutation, and silent mutations were respectively counted and summated, but germline mutation was excluded. Gene expression data from microarrays was normalized using a robust multiarray averaging method[66]. The BRB-ArrayTools software program (http://linus.nci.nih.gov/BRB-ArrayTools.html) was used to analyze gene expression data[67]. A heatmap was generated using the Cluster and TreeView software programs[68]. Other statistical analyses were performed in the R

language (http://www.r-project.org) or using STATA version 12 (StataCorp LP, College Station, TX, USA). To select genes that were differentially expressed between responder and non-responder of the training cohort (GSE35640)[24], we applied stringent cutoff of $P < 0.005$ (Student's t-test) and 1.5-fold difference and identified 105 genes. The signature was used to stratify patients in a validation cohort of GSE63557[28], human melanoma treated with anti-CTLA-4 antibody[29], and TCGA data. Of 105 genes of the training set, 27, 6, and 6 genes were excluded during validation with GSE63557 data, Van Allen et al. data, and TCGA data, respectively, due to difference in number of probes in microarray platforms or RNA-seq data (Supplementary Data 1). Gene expression data for the training and test sets were re-normalized by centralizing the gene expression level across the tissues. Briefly, expression data for 105 immune signature genes in the training set were combined to form a classifier according to a Bayesian compound covariate predictor (BCCP)[69]. The BCCP classifier estimated the likelihood that an individual patient had either a high immune signature or a low immune signature, according to a Bayesian probability of IS score cutoff of 0.5, which was optimized by comparing previously reported response rates to immunologic agents[7, 9–11] and the results of the current analysis of Receiver operating characteristic to predict the responder of a training cohort and a separate validation model by which cutoff is set by maximal point of sum of sensitivity and specificity[70]. To assess the degree of copy number variation which was calculated by Gistic 2.0[44], we defined "CIN score" as the summation of square of gene-level gistic 2 values. Adjusted CIN scores were computed by multiplying purity score (in range from 0 to 1) to original CIN scores.

**Canonical signaling pathways enriched in IS score**. Pathway analysis was carried by using Ingenuity Pathways Analysis and genes from the data set that were associated with a canonical pathway in the Ingenuity Pathways Knowledge Base were considered for the analysis. The significance of the association between immune signature and the canonical pathway was measured Fischer's exact test ($P < 0.001$). Among identified significant pathways, top 30 pathways were only reported in Supplementary Fig. 6. To estimate relative proportion of 22 types of infiltrated immune cells in tumor mass, online analytical platform CIBERSORT (https://cibersort.stanford.edu/) was used[35].

**Survival analysis**. Using IS score to dichotomize the patients into two subgroups (cutoff of 0.5), the prognostic significance was estimated using Kaplan–Meier plots (log-rank tests) and Cox proportional hazards regression analysis and then adjusted and stratified by cancer type. Prognostic significance of the continuous value of IS score was also calculated by Cox proportional hazards regression analysis. P-value < 0.05 was considered as a significant difference. Overall survival was gathered from TCGA clinical data, "days_to_last_follow-up" (CDE_ID: 3008273) if censored, or "days_to_death" (CDE_ID: 3165475) if dead. Progression-free survival (PFS) was measured from "days_to_drug_therapy_start" (CDE_ID: 3392465) until "days_to_drug_therapy_end" (CDE_ID: 3392470). PFS event was gathered from "therapy_ongoing" (CDE: 3103479).

**Significance of IS score according to genomic alterations**. The significance of global correlation between IS scores and number of mutations or CIN scores was estimated by linear regression analysis or generalized additive models (GAM) using R-Project statistical package. The significance of IS score difference according to clinicopathologic features such as the presence of virus, and mutation was estimated by Wilcox rank-sum test or analysis of variance (if more than three groups were compared). For each cancer type, we performed logistic analysis with IS score as the independent variable, and dichotomized status in genomic data such as higher or lower than median mutation number or CIN scores as the dependent variables. $P < 0.05$ was considered a significant difference.

To find specific mutations significantly associated with IS scores, Wilcoxon rank-sum tests were applied to the mean difference of IS score according to each mutation status (mutated versus wild-type). Likewise, significant difference of IS score by amplified or deleted genes were also identified by Wilcoxon rank-sum tests. To facilitate analysis, we limited analysis with previously recognized 373 driver genes[47] for mutation analysis and 87 amplified and 123 deleted genes[53] for CIN analysis. To estimate the significance of correlation in each cancer type, subgroup analysis of logistic regression was carried out to compute odds ratio (OR) of mutation rate or CIN score. False discovery rates were applied to control type I errors.

**Data availability**. The genomic data that support findings of this study are available from the NCBI Gene Expression Omnibus (GEO, http://www.ncbi.nlm.nih.gov/geo/) under accession number GSE35640, GSE63557, and GSE78220. Genomic data from TCGA project are available from the National Cancer Institute's Genomic Data Commons (https://gdc.cancer.gov/). All other data supporting the findings of this study are available within the article and its supplementary information files or from the corresponding author upon reasonable request.

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

## Acknowledgements

This study was supported in part by National Institutes of Health grant CA150229, CPRIT RP170307, UT M.D. Anderson Cancer Center 2016, Institutional Research Grant (IRG), and 2016 Sister Institute Network Fund (SINF). Additional support was provided by the National Institutes of Health through a Cancer Center Support Grant to The University of Texas MD Anderson Cancer Center (CA016672). We appreciated patients and their families who generously donated their tissues to TCGA, as well as the members of TCGA who collected and disclosed valuable data. Andre Kim foundation from Seoul National University Hospital, and Korean Association of Clinical Oncology as well as Dr. Do-Youn Oh, Dr. Seock-Ah Im, and Dr. Yung-Jue Bang from Seoul National University supported Dr. Chan-Young Ock for the training program of M.D. Anderson Cancer Center to study genomic analysis to perform this study. Dr. Se-Hoon Lee from Samsung Medical Center, Dr. Jisu Oh from Cha University, Dr. Choong-kun Lee from Yonsei University, Dr. Chi Young Ok from M.D. Anderson Cancer Center, Dr. Youngil Koh, Dr. Sehhoon Park, and Dr. Jonghanne Park from Seoul National University discussed about the concept of the current study.

## Author contributions

J.-S.L. designed and organized the experiment. C.-Y.O. designed and led the integrative analyses. C.-Y.O., J.-E.H., S.-B.K., J.-J.S., H.-J.J., S.P., B.H.S., M.C. and J.-S.L. performed analysis of genomic data. B.K., J.A.A., S.K., K.-W.L., T.M.K. and D.S.H. advised the comprehensive translation of the result. C.-Y.O. wrote the initial draft. B.K., J.A.A., S.K., K.-W.L., T.M.K. and J.-S.L. revised the draft. All authors read and approved the final manuscript.

## Additional information

**Competing interests:** The authors declare no competing financial interests.

