## [Peer Review File · Nature Communications]

Reviewers' comments:

Reviewer #1 (Remarks to the Author):

General comments

The manuscript by Ock et al. describes an immune gene signature for predicting response to immunotherapy and analysis of the TCGA data with respect to this signature. They show an association of the score that was developed with molecular subtypes of cancers. Furthermore, they show an association of the mutational burden (positive) and CNI (negative) with the score, and association with somatic mutations in 373 genes, copy number alterations, and viral presence.

There is an urgent need to identify predictive markers for selecting responders to therapy with checkpoint blockers and analyses of large-scale data could provide clues. However, the study has serious flaws and the presented results are very weak. Most importantly, treatment with cancer-germline antigen-based vaccination and checkpoint blockers have completely different mechanisms of actions and would be surprising that a signature identified from MAGE-A3 trial works also in a anti-CTLA-4 case. As a matter of fact, as shown in the manuscript in Figure 1, the performance of the signature is poor (AUC=0.7) and not significant ($p=0.2$) for the CTLA-4 study. Moreover, the authors did not perform comparative analyses of the proposed score with other markers suggested in the past.

Additionally, also the shown associations are weak (e.g. correlation with mutational load and CIN) and hence, do not provide additional insights. Calculations of correlations of a set of genes with other genomic features without strong results are not contributing to the knowledge gain.

Specific comments

1. The authors should elaborate on the 105 genes. It appears that only a fraction of them are expressed in immune cells (like CD8A, IL23A, CCL5) or related to immunity (like HLA, B2M). Although some immune-related pathways are shown in Figure S1, an in-depth analysis and discussion is necessary if this signature is supposed to be used in a clinical setting.
2. What is the performance of the signature compared to other proposed predictive markers like cytolytic activity, CTLA4 expression, PD-1/PD-L1 expression, neoantigen load, clonal origin of neoantigens or the like? The authors should carry out comparative analyses to provide evidence that the score they developed is indeed superior.
3. It does not make sense to analyze the TCGA data for potential responders and nonresponders without considering the stages. All checkpoint blockers are approved for late stage metastatic cancers and this has to be considered in the analysis.
4. The authors claim the score is correlated with interferon-gamma score and therefore is also a predictive marker for PD-1 therapy. This needs an independent validation, e.g. using the data from Hugo et al. (PMID:26997480).
5. A number of typos in the text and in the figures should be corrected.

Reviewer #2 (Remarks to the Author):

In this manuscript by Chan-Young Ock et al., the authors develop an Immune Score (IS) from gene expression predictor of immunotherapy response and applied this method to assess its prediction in genomic data from (N= ~10000) human tissue across 30 different cancer types (TCGA). The authors reports two distinct tumor types, i.e. Mutator type, which are associated with potential response to immunotherapy and chromosome instable type negatively associated with potential response to immunotherapy. In this study the authors use GSE35640 dataset to derive immune score and validated this signature on GSE63557 (mouse model dataset treated with anti-

CTLA4 antibody). In addition, the authors also validated the IS on RNASeq dataset from human melanoma treated with anti-CTLA4 antibody (Van Allen et al.). In summary the authors propose gene expression based "Immune score" in response to immunotherapy treatment.

Major Issues:

1. CTLA4 predominantly functions in the early stages of the immune response during T-Cell priming and activation and enhances immune suppressive activity of regulatory T-Cell while PD1-PDL1 receptor pair ligand is a dominant immune checkpoint pathway operative in tumor micro environment and functions in controlling immune homeostasis induced in cancer cells to evade immune attack. The distinct phenotype and functional properties of CTLA4 and PD1-PDL1 immune checkpoints enable the comparative assessment of gene expression signature. Since the authors specifically focus on the CTLA4 angle it will be more appropriate to change the title of the study from "...immunotherapy across cancer lineage" to "...immune score from anti-CTLA4 treated patient and its implication across cancer lineage".
2. The authors should discuss in more detail the samples showing high Immune Score with lower progression free survival.
3. Given that activation of specific oncogenic pathways can have broad effect on gene expression, it's reasonable to imagine that genetic makeup of a cancer cell could have a major effect on the tumor micro environment by driving a specific immune resistance pathway. Does this have any impact on the Immune Score prediction?
4. It would be interesting to see the difference in gene expression signature between "responders" vs "non-responders" from the MSI positive cases treated with a checkpoint blockade inhibitor.
5. As reported by the authors, the Immune Score derived from the dataset (GSE35640) was on patients who may have gone through multiple previous treatments which could alter the tumor microenvironment, hence the results in differential gene expression signature. Can the authors further elaborate on this point?
6. Can the authors describe the composition of tumor infiltrating leukocytes (% of macrophage, T-Cell, B-Cell, Neutrophils)? Are the abundance of specific leukocyte subsets associated with a specific gene signature in the tumor infiltrating leukocytes?
7. What are the determinants, such as stromal versus epithelial tumors, in each cancer type that dictate the involvement of immune cells to therapy response?
8. The authors should investigate the gene expression signature from the non-responder patients to identify any possible underlying mechanism toward resistance to immunotherapy.
9. The authors should also discuss the inflamed microenvironment vs non inflamed micro environment and their impact on the Immune Score.

Minor Issues:

1. Figure 1A "Responer" typographical error.
2. Figure Legend 2C states that "Of XXX patient of melanoma, only XX patients treated with immunotherapy were included in analysis". What is the number for the patients?
3. Figure Legend 4C states that "whereas the tumor with high chromosomal instability (C-type) is defined by chromosome instability score more than 5000". The authors should provide the details about the scoring method.

Reviewer #3 (Remarks to the Author):

The manuscript 'Genomic landscape associated with response to immunotherapy across cancer lineages' tackled the question whether host genetic characteristics dictate response to immunotherapy by analyzing genetic alteration and transcriptome changes from 10k cancer tissues from TCGA project spanning 30 different cancer types.

As surrogate of immune responsiveness, they compiled an immune gene signatures score (IS score) based on genes associated with responsiveness to immunotherapy. The IS scores were then correlated with qualitative and quantitative variations in term of copy number and somatic mutations and with other clinic pathological features. Key associations identified include those with mutational burden, copy number instability, and with mutations and copy number variations of specific genes involved in oncogenic process. Correlation between IS scores and molecular intrinsic subtypes is also a plus of this report. Interestingly, authors were able to define the proportion of samples, in each tumor type, that could potentially respond to immune manipulations. Quite interesting, this metrics in part recapitulate the response rate observed in clinical practice.

Although Authors used a pipeline that is similar to the one used by Rooney et al., which employed a cytolytic score (mean of GZMA and PRF gene) as target variable (Rooney et al. Cell, 2015 PMID: 25594174). A more robust immune signature score used in this study might have increased the sensitivity of such associative analyses and some results are novel and extremely interesting. Overall, this report contributes to increase the knowledge about the relationship between oncogenic process and spontaneous immune responsiveness and offers a useful resource for further immunogenomic analyses. The proposed classification should be further tested in immunotherapeutic trials.

While the manuscript is well written and the overall approach sounds, the following areas should be considered and revised:

- The classifier of 105 genes derived by the re-analysis of the MAGE-A3 vaccine study (Ulloa-Montoya, JCO 2013 PMID: 23715562) seems to almost perfectly overlap with the 100 differentially expressed gene originally identified by Ulloa-Montoya et al.. While this is somehow a proof that slightly different approaches are able to capture critical biological process, authors should consider to disclose the extent of the overlap between their classifier and the original one developed by Ulloa-Montoya et al. and reference the original publication.
- The authors' IS signature is centered on genes previously identified as the Immunologic Constant of Rejection (Wang E, Worschech A and Marincola FM – The immunological constant of rejection. Trends in Immunol – 29 (6):256-262, 2008; Wang E, et al. Antitumor vaccines, immunotherapy and the immunological constant of rejection. IDrugs 2009 12(5): 297-301; Wang E and Marincola FM – From the “delayed allergy reaction” to the “immunologic constant of rejection”; Wang E, Bedognetti D, Marincola FM. Prediction of Response to Anticancer Immunotherapy Using Gene Signatures. Journal of Clinical Oncology, Vol 31:1-3, 2013; Bedognetti D1, Hendrickx W, Marincola FM, Miller LD. Prognostic and predictive immune gene signatures in breast cancer. Curr Opin Oncol. 2015 Nov;27(6):433-44.) for example, CXCR3 and CCR5 ligand genes [CCL5, CXCL9, CXCL10], cytotoxic functions [eg PRF], and IFNG-signaling [eg, IRF1, STAT1, GBPs, HLAs transcripts] [Wang JCO 2013, PMID: 23715576; Galon Immunity 2013, PMID: 23890060]. This set of genes has been constantly found associated with increased survival in excised tumor (prognostic effect) and responsiveness to immunotherapeutic approaches (predictive effect) including anti-CTLA4 (Ji, CII 2011, PMID: 22146893), adoptive therapy/IL-2 (Weiss CCR 2011 PMID: 21976537, Bedognetti BJC 2013, PMID:24129241), vaccination (Ulloa-Montoya 2012 PMID:23715562) as well as with responsiveness to chemotherapy in breast cancer and with the

development of auto-immunity and allograft rejection. Author should better acknowledge previous studies and observations.

- For clarity, it would be useful to add in Table S1 which genes are up and which ones down regulated in Responders vs Non responders.
- Authors should fully describe how the IS score is computed. Authors mentioned to have used the Bayesian compound covariate predictor, but they reference Ramaswamy's paper, which instead describes support vector machines. No other details are provided. In the Bayesian compound covariate predictor (Radmacher MD & Simon R, J Comput Biol. 2002 PMID: 12162889), the t-statistic and log ratio of differentially expressed genes between class in comparisons is used as its weight in the compound covariate. Is the IS score in this analysis use the quantification of the compound covariate? How is the IS score of the TCGA samples computed? Was the t-statistic and log-ratio derived from the immunotherapy dataset used to weight the gene expression values? Is the IS score in the two validation dataset computed using the compound covariate predictor in such datasets or was it generated by applying the t-statistic metric of the training dataset? How were the RNA-seq processed and normalized? Clarification of this aspect is critical for the overall understanding and reproducibility of the results.
- It is unclear how the association between somatic mutations and IS score is performed. Authors stated that "Wilcoxon rank sum tests were applied to the mean difference of IS score between sensitive (IS score > 0.5) and resistant (IS score < 0.5) groups and number of mutations". This is quite difficult to understand, please elaborate and clarify, perhaps including examples.
- Line 399 "the number of mutation of IS score with somatic mutations and CIN score was calculated by logistic regression analysis". It is not possible to use logistic regression between continuous variables. I assume authors used regression-based approaches. Please clarify.
- Figure 2, Legend, "Kaplan-Meier plots of progression-free survival (PFS) of advanced melanoma patients in TCGA 678 treated with immunotherapy. Of XXX patients with melanoma, only XX patients treated with immunotherapy were included in the analysis". Please replace "XX" and "XXX" with the real numbers. In addition, from Table2B, it is evident that a significant proportion of patients received interferon as immunotherapy. It is likely that IFN was administered in the adjuvant rather than in the metastatic setting. If possible, please provide more information regarding these 82 patients: is the PFS calculated from the excision of the primary tumors or from the time of relapse? It should be taken into account that melanoma patients with only nodal disease receive surgery with curative intent are often treated with adjuvant IFN.
- The major funding authors reported is a strong negative correlation between copy number score "CIN" and IS score. Such correlation was much stronger than the one observed between IS score and somatic mutations. Although a flat copy number landscape in immune-enriched tumors has been described in some contexts (Curtis et al, Nature 2012 PMID: 22522925), authors here extend this observation in a pan-cancer fashion and using precise metrics. These results are indeed intriguing. Nevertheless, this negative correlation could be the result of a dilution effect exerted by the progressive enrichment of germline DNA in the immune-infiltrated tumors. Although, Authors anticipated this possibility and correlated the CIN score with the histologically assessed purity and found only modest association ($R_{sq} = 0.003$). However, the statement is not supported by figure S5 which shows obvious stronger correlation. Histologically assessed purity is more qualitative than quantitative. This issue has been systematically addressed in a recent work (Aran et al. Nat Commun 2015 PMID: 26634437). Correlations among genomic-based methods (ESTIMATE, ABSOLUT, and LUMP) were high while correlation with histologically assessed purity was low. It is possible that the scant association reported in the present report between purity and CIN is the result of the poor sensitivity of the H&E estimator. In addition, it is quite puzzling that authors reported a direct correlation between tumor purity and somatic mutations, while, in the same TCGA data set, a strong inverse correlation between somatic mutations and purity was found using

a robust purity estimator (CPE) based on the metrics of the 3 genomic estimators + the H&E score (R=-0.6) (Aran et al. Nat Commun 2015 PMID: 26634437). Overall it is not convincing (and unlikely) that that purity did not influence the negative correlation observed between CIN and IS score. Still, it is possible that the observed associations is biologically relevant. I encourage the authors to correlate both the IS scores and the CIN scores with the more robust CPE purity score. Single-patient CPE scores are available from Aran's paper. A conclusive demonstration of limited associations between CIN and purity is needed to support the strong statement made in the conclusion. Otherwise such intrinsic limitations need to be addressed in the discussion.

- It is not clear how the authors corrected the CIN for purity. Authors should make the CIN scores available; this could be added to TableS2.
- In the discussion part there is a strong emphasis on the associations between HLAs, CASP8, and M2B mutations and IS score. The same associations have been found previously by Rooney et al, and it should be acknowledged. Author should elaborate in discussion that this association with higher IS are not supported by the accumulated publications that HLA, B2M mutation or deletion in cancer are one of the immune escape mechanism.
- Line 357 material and method: Germline mutation/variants are inherited/carried by the tumor. Therefore the statement of " germline mutation without somatic mutation was excluded" is not clear statement. This sentence means that only somatic mutation were considered independent of germline mutation.
- Figure 4C exaggerated the data by using three way to emphasize of the positive immune score. The scale, the color and the size of the dots. The difference in size of the dots has misleading visual effect and should be avoid.

Reviewer #1 (Remarks to the Author):

General comments

The manuscript by Ock et al. describes an immune gene signature for predicting response to immunotherapy and analysis of the TCGA data with respect to this signature. They show an association of the score that was developed with molecular subtypes of cancers. Furthermore, they show an association of the mutational burden (positive) and CNI (negative) with the score, and association with somatic mutations in 373 genes, copy number alterations, and viral presence.

There is an urgent need to identify predictive markers for selecting responders to therapy with checkpoint blockers and analyses of large-scale data could provide clues. However, the study has serious flaws and the presented results are very weak. Most importantly, treatment with cancer-germline antigen-based vaccination and checkpoint blockers have completely different mechanisms of actions and would be surprising that a signature identified from MAGE-A3 trial works also in a anti-CTLA-4 case. As a matter of fact, as shown in the manuscript in Figure 1, the performance of the signature is poor (AUC=0.7) and not significant (p=0.2) for the CTLA-4 study. Moreover, the authors did not perform comparative analyses of the proposed score with other markers suggested in the past. Additionally, also the shown associations are weak (e.g. correlation with mutational load and CIN) and hence, do not provide additional insights. Calculations of correlations of a set of genes with other genomic features without strong results are not contributing to the knowledge gain.

> We thanks reviewer for pointing out few weakness of as well as constructive comments on our study. Association of our scores from prediction is not insignificant in clinical trial data as AUC is 0.7 and p-value is 0.02 (not 0.2) in **Figure 1F**. And this significance is well reflected in difference in overall survival and progression survival between patient groups with high and low IS scores (**Figure 1G and 1H**). As suggested, we included new analysis for comparison of our signature (or IS scores) with previously recognized potential predictor for immunotherapy, especially anti-PD1 antibody treatment.

We also used gene expression signature scores from recent KEYNOTE-012 trials. In this study, investigators found that gene scores of interferon gamma signature were highly associated with benefit of pembrolizumab (anti-PD1 antibody) treatments in head and neck cancer and gastric cancer (Lancet Oncology, PMID: 27157491; Lancet Oncology, PMID: 27247226). Correlation between IS scores and interferon gamma signature scores were highly significant (**Figure 2D**, $R^2=0.607$ and $P < 0.001$ in fact P-value is less than 10^{-22}). This strong correlation indicated that two different immunotherapy approaches (antigen-based vaccine and checkpoint blocker) share underlying biology of host immune cells responding to cancer cells. As our IS scores showed strong association with response to checkpoint inhibitors as well as strongly correlation with biomarker scores from checkpoint inhibitor treatment, we strongly believe that IS scores are highly comparable with data from checkpoint inhibitor treatment.

Specific comments

1. The authors should elaborate on the 105 genes. It appears that only a fraction of them are expressed in immune cells (like CD8A, IL23A, CCL5) or related to immunity (like HLA, B2M). Although some immune-related pathways are shown in Figure S1, an in-depth analysis and discussion is necessary if this signature is supposed to be used in a clinical setting.

> We thank reviewer for constructive comments. As suggested, we carried out additional analysis that can identify potential upstream regulators of 105 genes. In good agreement with pathway analysis, many of transcription regulators and cytokines known to be involved in immune related activities are identified as regulators of 105 genes and their activities are predicted to be high in responder subgroup. In addition, we also carried out analysis to identify potential regulators that are activated in non-responder and identified that MYC is highly active in non-responders, suggesting that MYC might be negative regulators of immune response. This observation is really in good agreement with recent study showing that MYC negatively regulates immune response (Casey et al., MYC regulates the antitumor immune response through CD47 and PD-L1. Science. 2016 352(6282):227-31. PMID: 26966191). Furthermore, in our independent analysis, MYC amplification was negatively associated with IS scores. We now included results of new analysis in **Table S2 and Figure S2**.

2. What is the performance of the signature compared to other proposed predictive markers like cytolytic activity, CTLA4 expression, PD-1/PD-L1 expression, neoantigen load, clonal origin of neoantigens or the like? The authors should carry out comparative analyses to provide evidence that the score they developed is indeed superior.

> As suggested, we carried out comparison with other proposed predictive markers. Since neoantigen load or clonal origin of neoantigens are currently not available from validation set, our analysis was limited to (1) Cytolytic activity (2) CTLA4, (3) PD1/PDL1 expression, and (4) interferon gamma signature. **(Figure S1A, B, C)**. All of analysis showed that IS scores are highly similar to or slightly better than previously recognized potential markers, suggesting that IS score is comparable with other markers.

3. It does not make sense to analyze the TCGA data for potential responders and nonresponders without considering the stages. All checkpoint blockers are approved for late stage metastatic cancers and this has to be considered in the analysis.

> We thank reviewer for thoughtful suggestion. However, use of TCGA data in our analysis is to find potential association of genomic traits of cancers with immune response. Furthermore, majority of TCGA tumors were collected in relatively earlier (pre-metastatic) stage in order to uncover primary driving genetic events during tumor development. Since majority of tumors are not in metastatic stage, we didn't carry out additional analysis as suggested.

4. The authors claim the score is correlated with interferon-gamma score and therefore is also a predictive marker for PD-1 therapy. This needs an independent validation, e.g. using the data from Hugo et al. (PMID:26997480).

> As suggested, we carried out additional analysis with new data set. However, while we found that there is association between two subtype and prognosis, it was weaker than what we observe in anti-CTLA-4 treatment cohort. This could be due to small sample size (only 28 patients) or may indicate some difference between anti-PD1 treatment and anti-CTLA4 treatment. We carried out additional analysis with other known markers such as cytolytic scores and interferon gamma signature scores, and found that other markers also showed lack of significant association, suggesting that small sample size might be blamed for weak association of these markers. (Please see reviewer's only material in supplementary data).

But, When we re-assessed TCGA data with available information on immunotherapy, mostly CTLA4 therapy **(Fig 2B)**, we found significant association of IS scores with prognosis. Taken together with strong

correlation with INF-gamma score and significant association in CTLA4 therapy, our study indicates that IS scores well reflect potential immune activity that is indicative of therapeutic benefit of patients.

5. A number of typos in the text and in the figures should be corrected.

> Thank you for pointing out our errors. We fixed them in revised manuscript.

Reviewer #2 (Remarks to the Author):

In this manuscript by Chan-Young Ock et al., the authors develop an Immune Score (IS) from gene expression predictor of immunotherapy response and applied this method to assess its prediction in genomic data from (N= ~10000) human tissue across 30 different cancer types (TCGA). The authors reports two distinct tumor types, i.e. Mutator type, which are associated with potential response to immunotherapy and chromosome instable type negatively associated with potential response to immunotherapy. In this study the authors use GSE35640 dataset to derive immune score and validated this signature on GSE63557 (mouse model dataset treated with anti-CTLA4 antibody). In addition, the authors also validated the IS on RNASeq dataset from human melanoma treated with anti-CTLA4 antibody (Van Allen et al.,). In summary the authors propose gene expression based "Immune score" in response to immunotherapy treatment.

Major Issues:

1. CTLA4 predominantly functions in the early stages of the immune response during T-Cell priming and activation and enhances immune suppressive activity of regulatory T-Cell while PD1-PDL1 receptor pair ligand is a dominant immune checkpoint pathway operative in tumor micro environment and functions in controlling immune homeostasis induced in cancer cells to evade immune attack. The distinct phenotype and functional properties of CTLA4 and PD1-PDL1 immune checkpoints enable the comparative assessment of gene expression signature. Since the authors specifically focus on the CTLA4 angle it will be more appropriate to change the title of the study from "...immunotherapy across cancer lineage" to "...immune score from anti-CTLA4 treated patient and its implication across cancer lineage".

> We thank reviewer for providing constructive comments and suggestion on our study. As correctly pointed out by reviewer, our signature is only validated in CTLA4 treated patients and underlying biology of CTLA4 immune checkpoint is different from PD1-PDL1 checkpoint. However, since our signature is not solely optimized for CTLA4 treatment (although showed good association), it might be pre-mature to call it as immune score for anti-CTLA4 treatment. Therefore, we did not change title in revised manuscript.

2. The authors should discuss in more detail the samples showing high Immune Score with lower progression free survival.

> As correctly pointed by reviewer, not all patients with high IS score had greater benefit of immunotherapy. In Figure 2C, median PFS of IS score high group is about 20 months, indicating that some (about 50%) patients with high IS scores has short PFS (less than 20 months). There would be other clinical factors affecting PFS of immunotherapy other than IS scores. Or it may indicate limitation of current IS scores. We included potential limitation of IS scores in discussion as suggested.

3. Given that activation of specific oncogenic pathways can have broad effect on gene expression, it's reasonable to imagine that genetic makeup of a cancer cell could have a major effect on the tumor micro environment by driving a specific immune resistance pathway. Does this have any impact on the Immune Score prediction?

> As correctly pointed out, we agree that some of genetic makeup (or genetic events in) of cancer cells would have substantial roles in immune resistance. To uncover potential players, we carried out association analysis of immune score with genetic events (Figure 6 and Figure 7). It identified amplification of MYC as potential resistant genetic event in multiple cancers. In addition, when we re-analyzed our immune signature, we also found that MYC's transcription activity was significantly increased in low immune score tumors (**Figure S2 and Table S2**). Our new findings is well supported by recent study showing that MYC is negative regulator of immune response (Casey et al., MYC regulates the antitumor immune response through CD47 and PD-L1. Science. 2016 352(6282):227-31. PMID: 26966191). In addition to MYC, we also uncovered several potential candidates that play roles in regulation of host immunity (summarized in Figure 6 and 7).

4. It would be interesting to see the difference in gene expression signature between "responders" vs "non-responders" from the MSI positive cases treated with a checkpoint blockade inhibitor.

> Since we don't have MSI data from patients treated with checkpoint blockade inhibitors, we simply used gene expression data from tumors with well-characterized MSI status (colorectal and stomach cancers) to find genes associated with high/low IS scores in MSI high patients. By applying stringent cut-off ($P < 0.005$) to find differentially expressed genes in high and low IS score tumors in MSI-H colorectal and stomach cancer, we identified 72 genes that are common in both cancer types (**Figure S6F**). Even though all MSI-H tumors have hyper-methylation and high mutation burdens, many genes involved in host immune activity are up-regulated in IS score-high tumors, suggesting that high mutation burden may not solely accountable for higher immune activity. While interesting, since in-depth analysis specific to MSI-H subtype would be beyond scope of current study, we did not carry out additional analysis. We included new analysis in revised manuscript.

5. As reported by the authors, the Immune Score derived from the dataset (GSE35640) was on patients who may have gone through multiple previous treatments which could alter the tumor microenvironment, hence the results in differential gene expression signature. Can the authors further elaborate on this point?

> We agree with reviewer that previous treatment may alter tumor microenvironment and thus have impact on gene expression signature. However, fortunately, patients in training set (GSE35640) were not previously treated according to eligibility criteria from original paper

Eligibility Criteria (from original JCO journal, PMID: 23715572) Patients had MAGE-A3-positive melanoma (stage III in-transit metastasis/unresectable stage III/stage IV-M1a) without prior treatment for metastases other than isolated limb perfusion.

We revised manuscript to include this information.

“Gene expression data from a randomized phase II trial of immunotherapy with MAGE-A3 antigen in malignant melanoma without prior treatment for metastases other than isolated limb perfusion were used for analysis^{21, 22}”

6. Can the authors describe the composition of tumor infiltrating leukocytes (% of macrophage, T-Cell, B-Cell, Neutrophils)? Are the abundance of specific leukocyte subsets associated with a specific gene signature in the tumor infiltrating leukocytes?

> As suggested, we carried out computational approach to estimate composition of immune cells in each tumor as well as average composition of cancer types by using newly developed tool from Alizadeh group: CIBERSORT (<https://cibersort.stanford.edu/>) (Nat Methods. 2015 May;12(5):453-7). This tool has been successfully used to estimate immune cell composition in many previous studies (Nat Med. 2015. 21(8):938-45; J Natl Cancer Inst. 2016. 109(1); PLoS Med. 2016. 13(12):e1002194). Not surprisingly, CD8+ T cells and M1 macrophages showed high percentage among immune cells. Furthermore, they are significantly associated with IS scores. New analyses are now included in revised manuscript (**Figure S4**).

7. What are the determinants, such as stromal versus epithelial tumors, in each cancer type that dictate the involvement of immune cells to therapy response?

> To estimate stromal effects on ISS, we used IHC information from TCGA pathology report that measured percentage of cancer cells in tumor mass. Our ISS is positively correlated with fraction of stromal cells (**Fig. S7C**). To further confirm this, we used previously developed genomic level purity scores that were established by Atul Buttes group (Aran D et al., Systematic pan-cancer analysis of tumour purity. Nat Commun. 2015. 6:8971). They first generated three different purity scores by using different genomic data: ESTIMATE, which uses gene expression profiles of stromal genes; ABSOLUTE, which uses somatic copy-number data; LUMP, which averages 44 non-methylated immune-specific CpG sites. Consensus measurement of purity estimations (CPE) was later generated to represent overall purity of tumor mass by integrating all three different purity scores. Our ISS is negatively correlated with CPE scores (representing tumor percentage) (**Fig S7C**), suggesting that higher stromal compartment in tumor mass indicate higher infiltration of immune cells to tumor mass and thus contribute more to high immune activity. We now include new analysis in revised manuscript.

8. The authors should investigate the gene expression signature from the non-responder patients to identify any possible underlying mechanism toward resistance to immunotherapy.

> As suggested, we carried out gene network analysis with genes in immune signature. New analysis revealed several potential regulators of genes specific for non-responder patients (**Table S2 and Figure S2**). As expected, predicted active regulators include IL10 and its receptor IL10RA that are best known for anti-inflammatory cytokine/receptor. Furthermore, SOCS1 and SOCS3, negative regulators of cytokine signaling, are also active upstream regulators in non-responders, suggesting that de-regulated genes well reflect immunological characteristics in non-responder patients. Analysis also revealed MYC as potential active regulator in non-responders. This is in good agreement with our independent analysis showing that amplification of MYC is potential immune resistant genetic event in multiple cancers. This association is well supported by recent study showing that MYC is negative regulator of immune response (Casey et al., MYC regulates the antitumor immune response through CD47 and PD-L1).

Science. 2016 352(6282):227-31. PMID: 26966191). In addition to these findings, we also found several potential negative regulators that are summarized in figure 6 and 7.

9. The authors should also discuss the inflamed microenvironment vs non inflamed micro environment and their impact on the Immune Score.

> Since inflamed microenvironment would be the result of recruitment of inflammatory cells including anti-tumor immune cells, our data suggest that IS scores well reflect inflamed microenvironment. We now included this idea in revised discussion.

Minor Issues:

1. Figure 1A “Responer” typographical error.

> Thank you for pointing out our errors. We corrected it.

2. Figure Legend 2C states that “Of XXX patient of melanoma, only XX patients treated with immunotherapy were included in analysis”. What is the number for the patients?

> We corrected it.

3. Figure Legend 4C states that “whereas the tumor with high chromosomal instability (C-type) is defined by chromosome instability score more than 5000”. The authors should provide the details about the scoring method.

> CIN score was defined by sum of square of gene-level gistic 2 value. We described in methods section but may not be obviously visible to readers. To improve visibility of description, we include definition of CIN core in figure legend as well.

Reviewer #3 (Remarks to the Author):

The manuscript ‘Genomic landscape associated with response to immunotherapy across cancer lineages’ tackled the question whether host genetic characteristics dictate response to immunotherapy by analyzing genetic alteration and transcriptome changes from 10k cancer tissues from TCGA project spanning 30 different cancer types.

As surrogate of immune responsiveness, they compiled an immune gene signatures score (IS score) based on genes associated with responsiveness to immunotherapy. The IS scores were then correlated with qualitative and quantitative variations in term of copy number and somatic mutations and with other clinic pathological features. Key associations identified include those with mutational burden, copy number instability, and with mutations and copy number variations of specific genes involved in oncogenic process. Correlation between IS scores and molecular intrinsic subtypes is also a plus of this report. Interestingly, authors were able to define the proportion of samples, in each tumor type, that could potentially respond to immune manipulations. Quite interesting, this metrics in part recapitulate the response rate observed in clinical practice.

Although Authors used a pipeline that is similar to the one used by Rooney et al., which employed a cytolytic score (mean of GZMA and PRF gene) as target variable (Rooney et al. Cell, 2015 PMID: 25594174). A more robust immune signature score used in this study might have increased the sensitivity of such associative analyses and some results are novel and extremely interesting. Overall, this report contributes to increase the knowledge about the relationship between oncogenic process and spontaneous immune responsiveness and offers a useful resource for further immunogenomic analyses. The proposed classification should be further tested in immunotherapeutic trials.

While the manuscript is well written and the overall approach sounds, the following areas should be considered and revised:

- The classifier of 105 genes derived by the re-analysis of the MAGE-A3 vaccine study (Ulloa-Montoya, JCO 2013 PMID: 23715562) seems to almost perfectly overlap with the 100 differentially expressed gene originally identified by Ulloa-Montoya et al.. While this is somehow a proof that slightly different approaches are able to capture critical biological process, authors should consider to disclose the extent of the overlap between their classifier and the original one developed by Ulloa-Montoya et al. and reference the original publication.

> Ulloa-Montoya et al identified 100 gene features (probes) representing 84 genes by applying signal-to-noise (s2n) score approaches. Subsequently, they used supervised principal component-discriminant analysis (SPC-DA) classifier to test robustness of signature and achieved 0.77 of sensitivity and 0.56 of specificity. By using slightly different statistical analysis (Student t-test), we identified 105 unique genes and use them to develop classifier (Bayesian compound covariate predictor) to generate Immune Signature Score (ISS). While sensitivity (0.706) is slight lower, specificity (0.818) is better than original predictor. When we compared list of genes in two predictors, we found that substantial number of genes were overlapped (49 genes). We included this information in revised Table S1.

- The authors' IS signature is centered on genes previously identified as the Immunologic Constant of Rejection (Wang E, Worschech A and Marincola FM – The immunological constant of rejection. Trends in Immunol – 29 (6):256-262, 2008; Wang E, et al. Antitumor vaccines, immunotherapy and the immunological constant of rejection. IDrugs 2009 12(5): 297-301; Wang E and Marincola FM – From the “delayed allergy reaction” to the “immunologic constant of rejection”; Wang E, Bedognetti D, Marincola FM. Prediction of Response to Anticancer Immunotherapy Using Gene Signatures. Journal of Clinical Oncology, Vol 31:1-3, 2013; Bedognetti D1, Hendrickx W, Marincola FM, Miller LD. Prognostic and predictive immune gene signatures in breast cancer. Curr Opin Oncol. 2015 Nov;27(6):433-44.) for example, CXCR3 and CCR5 ligand genes [CCL5, CXCL9, CXCL10], cytotoxic functions [eg PRF], and IFNG-signaling [eg, IRF1, STAT1, GBPs, HLAs transcripts] [Wang JCO 2013, PMID: 23715576; Galon Immunity 2013, PMID: 23890060]. This set of genes has been constantly found associated with increased survival in excised tumor (prognostic effect) and responsiveness to immunotherapeutic approaches (predictive effect) including anti-CTLA4 (Ji, CII 2011, PMID: 22146893), adoptive therapy/IL-2 (Weiss CCR 2011 PMID: 21976537, Bedognetti BJC 2013, PMID:24129241), vaccination (Ulloa-Montoya 2012 PMID:23715562) as well as with responsiveness to chemotherapy in breast cancer and with the development of auto-immunity and allograft rejection. Author should better acknowledge previous studies and observations.

> add reference as indicated in manuscript.

- For clarity, it would be useful to add in Table S1 which genes are up and which ones down regulated in Responders vs Non responders.

> As suggested, we included regulation patterns of gene expression in revised **Table S1**.

- Authors should fully describe how the IS score is computed. Authors mentioned to have used the Bayesian compound covariate predictor, but they reference Ramaswamy's paper, which instead describes support vector machines. No other details are provided. In the Bayesian compound covariate predictor (Radmacher MD & Simon R, J Comput Biol. 2002 PMID: 12162889), the t-statistic and log ratio of differentially expressed genes between class in comparisons is used as its weight in the compound covariate. Is the IS score in this analysis use the quantification of the compound covariate? How is the IS score of the TCGA samples computed? Was the t-statistic and log-ratio derived from the immunotherapy dataset used to weight the gene expression values? Is the IS score in the two validation dataset computed using the compound covariate predictor in such datasets or was it generated by applying the t-statistic metric of the training dataset? How were the RNA-seq processed and normalized? Clarification of this aspect is critical for the overall understanding and reproducibility of the results.

> Thank you for pointing out our errors in manuscript. We incorporated incorrect reference in manuscript. Correct reference for BCCP should be "Wright G et al., A gene expression-based method to diagnose clinically distinct subgroups of diffuse large B cell lymphoma. Proc Natl Acad Sci U S A. 2003 Aug 19;100(17):9991-6". The predictor was later fine-tuned by Richard Simon's group at NCI and was incorporated to BRB-Arraytools (Simon R et al., Analysis of gene expression data using BRB-ArrayTools. Cancer informatics 3, 11-17, 2007). Our group have been extensively used this tool for many studies and described details of BCCP in previous publications (Oh et al., Prognostic gene expression signature associated with two molecularly distinct subtypes of colorectal cancer. Gut. 2012 61(9):1291-8; Sohn BH et al., Inactivation of Hippo Pathway Is Significantly Associated with Poor Prognosis in Hepatocellular Carcinoma. Clin Cancer Res. 2016 Mar 1;22(5):1256-64). Because its details were well described in these two publications, we did not include it in revised manuscript. But, it was included in revised manuscript as reviewer's only material for convenience. We also included details on data normalization in methods.

- It is unclear how the association between somatic mutations and IS score is performed. Authors stated that "Wilcoxon rank sum tests were applied to the mean difference of IS score between sensitive (IS score > 0.5) and resistant (IS score < 0.5) groups and number of mutations". This is quite difficult to understand, please elaborate and clarify, perhaps including examples.

> To improved clarity of our methods on finding association, we modified description as below in Methods.

"To find specific mutations significantly associated with IS scores, Wilcoxon rank sum tests were applied to the mean difference of IS score according to each mutation status (mutated versus wild type). Likewise, significant difference of IS score by amplified or deleted genes were also identified by Wilcoxon rank sum tests."

- Line 399 "the number of mutation of IS score with somatic mutations and CIN score was calculated by logistic regression analysis". It is not possible to use logistic regression between continuous variables. I assume authors used regression-based approaches. Please clarify.

> Thank you for pointing out ambiguity of our description on methods. To improve clarity of description,

we re-stated to “For each cancer type, we performed logistic analysis with IS score as the independent variable, and dichotomized status in genomic data such as higher or lower than median mutation number or CIN scores as the dependent variables.”

- Figure 2, Legend, “Kaplan-Meier plots of progression-free survival (PFS) of advanced melanoma patients in TCGA 678 treated with immunotherapy. Of XXX patients with melanoma, only XX patients treated with immunotherapy were included in the analysis”. Please replace “XX” and “XXX” with the real numbers. In addition, from Table2B, it is evident that a significant proportion of patients received interferon as immunotherapy. It is likely that IFN was administered in the adjuvant rather than in the metastatic setting. If possible, please provide more information regarding these 82 patients: is the PFS calculated from the excision of the primary tumors or from the time of relapse? It should be taken into account that melanoma patients with only nodal disease receive surgery with curative intent are often treated with adjuvant IFN.

> We included number of patients as suggested. While revisiting clinical data of TCGA, we found that three patients received treatment as adjuvant therapy, we removed these patients from analysis. Thus, total number of patients received immunotherapy is 78, not 81. We now corrected number of patients in revised manuscript. Re-analysis of PFS data with 78 patients did not change prognostic significance. Since clinical data of TCGA is not complete as much as data from clinical trials, we don't have information on if patient treated IFN-alpha as adjuvant therapy.

Relevant changes in revised manuscript are following.

1. *In figure legend, Of 472 patients with melanoma, only 78 patients treated with immunotherapy were included in analysis.*
2. *Figure 2B changed*
3. *Method*
Of 472 patients with skin cutaneous melanoma, 78 patients treated with immunotherapy which purpose is not indicated by "adjuvant" with appropriately annotated survival data were included in progression-free survival analysis.
4. *Overall survival was gathered from TCGA clinical data, "days_to_last_followup" (CDE_ID: 3008273) if censored, or "days_to_death" (CDE_ID: 3165475) if dead. Progression-free survival (PFS) was measured from "days_to_drug_therapy_start" (CDE_ID: 3392465) until "days_to_drug_therapy_end" (CDE_ID: 3392470). PFS event was gathered from "therapy_ongoing" (CDE: 3103479).*

- The major funding authors reported is a strong negative correlation between copy number score “CIN” and IS score. Such correlation was much stronger than the one observed between IS score and somatic mutations. Although a flat copy number landscape in immune-enriched tumors has been described in some contexts (Curtis et al, Nature 2012 PMID: 22522925), authors here extend this observation in a pan-cancer fashion and using precise metrics. These results are indeed intriguing. Nevertheless, this negative correlation could be the result of a dilution effect exerted by the progressive enrichment of germline DNA in the immune-infiltrated tumors. Although, Authors anticipated this possibility and

correlated the CIN score with the histologically assessed purity and found only modest association ($R_{sq} = 0.003$). However, the statement is not supported by figure S5 which shows obvious stronger correlation. Histologically assessed purity is more qualitative than quantitative. This issue has been systematically addressed in a recent work (Aran et al. Nat Commun 2015 PMID: 26634437). Correlations among genomic-based methods (ESTIMATE, ABSOLUT, and LUMP) were high while correlation with histologically assessed purity was low. It is possible that the scant association reported in the present report between purity and CIN is the result of the poor sensitivity of the H&E estimator. In addition, it is quite puzzling that authors reported a direct correlation between tumor purity and somatic mutations, while, in the same TCGA data set, a strong inverse correlation between somatic mutations and purity was found using a robust purity estimator (CPE) based on the metrics of the 3 genomic estimators + the H&E score ($R = -0.6$) (Aran et al. Nat Commun 2015 PMID: 26634437). Overall it is not convincing (and unlikely) that that purity did not influence the negative correlation observed between CIN and IS score. Still, it is possible that the observed associations is biologically relevant. I encourage the authors to correlate both the IS scores and the CIN scores with the more robust CPE purity score. Single-patient CPE scores are available from Aran's paper. A conclusive demonstration of limited associations between CIN and purity is needed to support the strong statement made in the conclusion. Otherwise such intrinsic limitations need to be addressed in the discussion.

> We thank reviewer for sharing great insights on complicated tumor purity issues and suggestion to strengthen our analysis and relevant conclusions. As suggested, we used Aran's CPE score to re-adjust CIN score. As expected, CIN score showed weak positive correlation with CPE as seen in IHC-based tumor purity data (slightly higher though). However, we still observed strong negative correlation between ISS and re-adjusted CIN scores (adjusted by CPE), suggesting that tumor purity has limited influence on observed negative correlation. We include new analysis in revised manuscript (Figure S7B) and include potential limitation of analysis in revised discussion.

- It is not clear how the authors corrected the CIN for purity. Authors should make the CIN scores available; this could be added to Table S2.

> We used simplest approach for adjustment by multiplying purity score (0 to 1 scale) to original CIN score. We included this description in revised manuscript. CIN score is now included in revised Table S2.

- In the discussion part there is a strong emphasis on the associations between HLAs, CASP8, and M2B mutations and IS score. The same associations have been found previously by Rooney et al, and it should be acknowledged. Author should elaborate in discussion that this association with higher IS are not supported by the accumulated publications that HLA, B2M mutation or deletion in cancer are one of the immune escape mechanism.

> We cited Rooney's paper and stated limitation of our findings in discussion.

- Line 357 material and method: Germline mutation/variants are inherited/carried by the tumor. Therefore the statement of "germline mutation without somatic mutation was excluded" is not clear statement. This sentence means that only somatic mutation were considered independent of germline mutation.

> We corrected as following, "*but germline mutation was excluded*"

- Figure 4C exaggerated the data by using three way to emphasize of the positive immune score. The scale, the color and the size of the dots. The difference in size of the dots has misleading visual effect

and should be avoid.

> We corrected as suggested.

Reviewers' comments:

Reviewer #1 (Remarks to the Author):

The authors addressed all issues raised in the original review, albeit the results are not completely convincing. The additional analyses that were carried out support the major finding that the proposed immune signature predicts response to immunotherapy with anti-CTLA-4 antibodies. However, if not in the title (as suggested by Reviewer 2) the authors should be more specific at least in the abstract and make it clear that the score is predicting response to treatment with anti-CTLA-4 antibodies. Furthermore, although negative, the results of the analysis of the Hugo data set are of interest to the readers and should be provided in the supplementary material.

Reviewer #2 (Remarks to the Author):

The authors have revised the manuscript addressing and incorporating some changes suggested by the reviewers, however additional analysis is required to clarify the points listed below.

Major Issues:

1. The title of the paper is misleading. The authors should not generalize across cancer lineage as the response data is only available from the melanoma study.
2. The authors have the data from MAGE-A3 (N=65) and CTLA-4 immunotherapy from 2 different studies (GSE63557 mouse model dataset and RNAseq dataset from human melanoma treated with anti-CTLA4 (N=78) antibody (Van Allen et al.)). The immunotherapy has different response mechanisms behind them so the robustness of the immune signature score should be validated on the immunotherapy treated patients from different cancer cohorts.
3. The study has limitations as the gene signature that the authors came up with is from a melanoma cohort treated with MAGE-A3 and anti-CTLA4, but the authors do not have samples treated with anti-CTLA4 from across other cancer types in spite of the differences in genetic makeup of the cancer as well as tumor micro environment.
4. The gene expression based classifier is basically built from the melanoma patients which has more homogeneous data. The author should be aware of the biopsy sites, tumor contents and cancer type. All these factor have a huge influence on gene expression along with no representation of treated samples from other cancer subtypes. This is a major weakness of the classifier built for the immune signature score.

Reviewer #3 (Remarks to the Author):

The authors promptly revised this interesting manuscript and assessed all our concerns except one, which we believe it is critical. This is about the relationship between CIN and IS score. Following the new analyses, it is clear that 1) purity (CPE) correlates strongly with IS score (please provide Rsq and P values for this panel and for all the other scatter plots) 2) CIN correlates with CPE ($R_s = 0,083$ and $P=10E-6$). Authors emphasize the correlation between CIN and IS score, and at the same time minimize the relevance of the correlation between CIN and CPE purity ("...strongly suggesting that CIN might be a more important predictor of clinical outcomes of immunotherapy than mutation rates"... "correlation between CIN scores and tumor purity [CPE] purity was modest". Such statements are misleading and not supported by the data, as the degree of the correlation between i) CIN and CPE and ii) CIN and IS is similar (being, according to the

authors, $R_{sq} = 0.083$, and $R_{sq} = 0.095$), which it is a logical consequence of the existence of a strong correlation between IS and CPE. If one compare scatter plots of i) CIN vs CPE (Fig s7B) and ii) CIN vs IS (Fig 6B) it is clear that the degree of the two correlations is nearly identical, suggesting therefore that the correlation between CIN and IS is strongly influenced by purity (please correct the dotted regression line in Fig6B: the slope of such line does not reflect the correlation index: from such regression line correlation between CIN and IS score seems about 0.6-0.8 while it is just 0.095.)

More in detail:

-Regarding the correlation between IS and CIN, and CIN and histologically-assessed purity, the authors wrote: " we estimated the impact of tumor purity in our analysis by examining the correlation of CIN scores with histologically assessed tumor purity. The correlation between CIN scores and tumor purity was only modest (Figure S7A)[...]Furthermore, the significance is not markedly altered by reanalysis of integrated data with adjusted CIN scores (Figure S7A, bottom), strongly indicating a minimum impact of tumor purity in our analysis."

-Regarding the correlation between IS and CIN, and CIN and "genomic-assessed" purity, the authors wrote "To further validate insignificant contribution of tumor purity to CIN and IS scores, we adopted previously established genomic approach, consensus measurement of purity estimations (CPE), for estimation of tumor purity that use gene expression, copy number alterations, and methylation data. As seen with IHC data, the correlation between CIN scores and tumor purity was modest (Figure S7B, top) and the significance is not markedly altered by reanalysis of integrated data with adjusted CIN scores (Figure S7B,)." .

So, what Figure S7A and figure S7B should show is the persistency of the correlation between, respectively, a) IHC-purity adjusted CIN and IS score (Fig S7A) and b) CPE-purity adjusted CIN and IS score. (Fig S7B). Neither Fig S7A nor Fig S7B show such correlations. Please provide the R_{sq} and P values of a) and b).

In conclusion, the authors should provide more convincing evidence of the "purity-independent" correlation between IS and CIN or e revise the manuscript accordingly.

Reviewer #1 (Remarks to the Author):

The authors addressed all issues raised in the original review, albeit the results are not completely convincing. The additional analyses that were carried out support the major finding that the proposed immune signature predicts response to immunotherapy with anti-CTLA-4 antibodies. However, if not in the title (as suggested by Reviewer 2) the authors should be more specific at least in the abstract and make it clear that the score is predicting response to treatment with anti-CTLA-4 antibodies. Furthermore, although negative, the results of the analysis of the Hugo data set are of interest to the readers and should be provided in the supplementary material.

> We now included analysis of Hugo data in Supplementary Figure 1D. As mentioned in previous version of manuscript, our IS score was not significantly associated with response to blockade of PD-1 checkpoint inhibitor. However, it should be noted that all other known predictive markers such as interferon gamma signature, cytolytic score, and expression of PD-1, PD-L1, and CTLA4 are also not well associated with response to treatment in this data set, suggesting that lack of association might be due to small sample size.

Reviewer #2 (Remarks to the Author):

The authors have revised the manuscript addressing and incorporating some changes suggested by the reviewers, however additional analysis is required to clarify the points listed below.

Major Issues:

1. The title of the paper is misleading. The authors should not generalize across cancer lineage as the response data is only available from the melanoma study.

> As suggested by reviewer #1 and #2, we changed title to “Genomic landscape associated with potential response to anti-CTLA4 treatment in cancers”.

2. The authors have the data from MAGE-A3 (N=65) and CTLA-4 immunotherapy from 2 different studies (GSE63557 mouse model dataset and RNAseq dataset from human melanoma treated with anti-CTLA4 (N=78) antibody (Van Allen et al.,). The immunotherapy has different response mechanisms behind them so the robustness of the immune signature score should be validated on the immunotherapy treated patients from different cancer cohorts.

> As suggested, we tried to find genomic data set from non-melanoma patients treated with blockade of immune checkpoint inhibitors. However, genomics in immunotherapy is still in early stage, we were not able to find good genomic data with good follow-up data, especially for different cancer cohorts. Only data from other cancer types were those from renal cell carcinoma although it only contains 11 patients treated with nivolumab (GSE67501). Similar to Hugo’s data, our predictor as well as other predictors are not well associated with response to treatment. As correctly pointed out by reviewer, it may be due to limitation of our predictor in finding responders to anti-PD-1 therapy. We included this new data in Supp Figure 1E and emphasize limitation of our genomic predictor in discussion as suggested.

(page 6, line 3)

IS score was not well associated with response to treatment with anti-PD-1 antibody in melanoma (N = 27) and renal cell carcinoma (N = 10), suggesting potential limitation of IS score predicting response to different immunotherapies (Figure S1D and S1E). However, it is worthwhile to point out that all other immune biomarkers failed to identify responders in these cohorts, indicating that lack of association might be due to small sample size.

3. The study has limitations as the gene signature that the authors came up with is from a melanoma cohort treated with MAGE-A3 and anti-CTLA4, but the authors do not have samples treated with anti-CTLA4 from across other cancer types in spite of the differences in genetic makeup of the cancer as well as tumor micro environment.

> We fully agree with reviewer on this. However, as explained in earlier, we were not able to find good data set to test this. Therefore, we changed title of manuscript and emphasize the limitation of current study in discussion.

(page 15, line 19)

Moreover, our result should be interpreted carefully when it applied to other cancer types as IS score is mostly validated in melanoma. Differences in genetic makeup of cancer cells and tumor microenvironment might have substantial influence on IS score in other cancer types. This should be further tested and validated in future studies with data from prospectively collected samples.

4. The gene expression based classifier is basically built from the melanoma patients which has more homogeneous data. The author should be aware of the biopsy sites, tumor contents and cancer type. All these factor have a huge influence on gene expression along with no representation of treated samples from other cancer subtypes. This is a major weakness of the classifier built for the immune signature score.

> Thank you very much for very insightful thoughts on this issue. As suggested, we included limitation of study in discussion.

Reviewer #3 (Remarks to the Author):

The authors promptly revised this interesting manuscript and assessed all our concerns except one, which we believe it is critical. This is about the relationship between CIN and IS score.

Following the new analyses, it is clear that 1) purity (CPE) correlates strongly with IS score (please provide Rsq and P values for this panel and for all the other scatter plots)

> As suggested, we included Rsq and P values in Supplementary Figure 7C.

2) CIN correlates with CPE ($R_s = 0,083$ and $P=10E-6$). Authors emphasize the correlation between CIN and IS score, and at the same time minimize the relevance of the correlation between CIN and CPE purity (“...strongly suggesting that CIN might be a more important predictor of clinical outcomes of immunotherapy than mutation rates”.... “correlation between CIN scores and tumor purity [CPE] purity was modest”. Such statements are misleading and not supported by the data, as the degree of the correlation between i) CIN and CPE and ii) CIN and IS is similar (being, according to the authors, $R_{sq} = 0.083$, and $R_{sq}=0.095$), which it is a logical consequence of the existence of a strong correlation between IS and CPE.

If one compare scatter plots of i) CIN vs CPE (Fig s7B) and ii) CIN vs IS (Fig 6B) it is clear that the degree of the two correlations is nearly identical, suggesting therefore that the correlation between CIN and IS is strongly influenced by purity (please correct the dotted regression line in Fig6B: the slope of such line does not reflect the correlation index: from such regression line correlation between CIN and iS score seems about 0.6-0.8 while it is just 0.095.)

> we corrected regression slope in Figure 6B as suggested.

More in detail:

-Regarding the correlation between IS and CIN, and CIN and histologically-assessed purity, the authors wrote: “ we estimated the impact of tumor purity in our analysis by examining the correlation of CIN scores with histologically assessed tumor purity. The correlation between CIN scores and tumor purity was only modest (Figure S7A)[....]Furthermore, the significance is not markedly altered by reanalysis of integrated data with adjusted CIN scores (Figure S7A, bottom), strongly indicating a minimum impact of tumor purity in our analysis.”

-Regarding the correlation between IS and CIN, and CIN and “genomic-assessed” purity, the authors wrote “To further validate insignificant contribution of tumor purity to CIN and IS scores, we adopted previously established genomic approach, consensus measurement of purity estimations (CPE), for estimation of tumor purity that use gene expression, copy number alterations, and methylation data. As seen with IHC data, the correlation between CIN scores and tumor purity was modest (Figure S7B, top) and the significance is not markedly altered by reanalysis of integrated data with adjusted CIN scores (Figure S7B,)”.

So, what Figure S7A and figure S7B should show is the persistency of the correlation between, respectively, a) IHC-purity adjusted CIN and IS score (Fig S7A) and b) CPE-purity adjusted CIN and IS score. (Fig S7B). Neither Fig S7A nor Fig S7B show such correlations. Please provide the Rsq and P values of a) and b).

In conclusion, the authors should provide more convincing evidence of the “purity-independent” correlation between IS and CIN or e revise the manuscript accordingly.

> As correctly pointed out by reviewer, correlation between IS and CIN appears to be modest in given Rsq value (0.095). However, it is also important to point out that Rsq value of correlation between

mutation burden and IS (well recognized association that is validated multiple times in previous studies, Supple Figure S6B top panel) is only 0.017. Correlation with CIN is higher than one with mutation burden. Furthermore, our finding is further supported by independent studies. Recent study from Stephen Elledge group (Tumor aneuploidy correlates with markers of immune evasion and with reduced response to immunotherapy, Science. 2017 Jan 20;355, issue6322, eaaf8399) showed strong correlation between copy number alteration and response to immunotherapy. In good agreement with our results, they found that copy number alteration is negatively correlated with response to immunotherapy. In analysis of samples from clinical trials with CTLA-4 and PD-1 blockade treatment, Roh et al also found that copy number loss is associated with resistance to immunotherapy (Integrated molecular analysis of tumor biopsies on sequential CTLA-4 and PD-1 blockade reveals markers of response and resistance (Sci Transl Med. 2017 Mar 1;9(379). pii: eaah3560). We included these new studies in revised discussion. In addition, we added new plots by IHC-adjusted CIN vs. ISS (Figure S7A), and CPE-adjusted CIN vs. ISS (Figure S7B). Adjusted CIN scores were still significantly associated with IS.

REVIEWERS' COMMENTS:

Reviewer #2 (Remarks to the Author):

The authors have revised the manuscript addressing and incorporating most of the changes suggested by the reviewers however there are some minor issues.

1. The authors changed the title but in the main text they are still using the generic term immunotherapies/immunotherapy approaches instead of CTLA-4 or MAGE-A3 therapy. (ie. line 98 "(IS score), is associated with response to different immunotherapy approaches including antigen based immunotherapy and immune checkpoint inhibitors", line 120 "Having found that the IS score reflected response to immunotherapies" and so on).

2. The authors removed KICH from the analysis but KICH have more chromosomal level losses and gains which might be a good cohort to see the correlation between CIN and IS scores.

3. The Supplemental figure legends still have the old title.

Reviewer #3 (Remarks to the Author):

The authors have address all the comments precisely and accurately. it is now ready to be published.